# Beyond the Final Layer: Intermediate Representations for Better Multilingual Calibration in Large Language Models

## Abstract

Confidence calibration, the alignment of a model's predicted confidence with its actual accuracy, is crucial for the reliable deployment of Large Language Models (LLMs). However, this critical property remains largely under-explored in multilingual contexts. In this work, we conduct the first large-scale, systematic studies of multilingual calibration across six model families and over 100 languages, revealing that non-English languages suffer from systematically worse calibration. To diagnose this, we investigate the model's internal representations and find that the final layer, biased by English-centric training, provides a poor signal for multilingual confidence. In contrast, our layer-wise analysis uncovers a key insight that late-intermediate layers consistently offer a more reliable and better-calibrated signal. Building on this, we introduce a suite of training-free methods, including Language-Aware Confidence Ensemble (LACE), which adaptively selects an optimal ensemble of layers for each specific language. Our study highlights the hidden costs of English-centric alignment and offer a new path toward building more globally equitable and trustworthy LLMs by looking beyond the final layer.

## 1 Introduction

Calibration in machine learning denotes the alignment between a model's predicted confidence and the empirical probability that its predictions are correct (Guo et al., 2017; Geng et al., 2024; Zhang et al., 2025a;b).[1] A model is perfectly calibrated if predictions assigned 80% confidence are correct approximately 80% of the time. Calibration is particularly critical for large language models (LLMs) in high-stakes applications such as medical diagnosis, legal advice, and decision support, where miscalibration can amplify harm (Zhang et al., 2024b; Yang et al., 2024b; 2025). Well-calibrated LLMs make their reliability explicit and interpretable, improving downstream trust and safety.

Despite its importance, most calibration work in LLMs focused on English (Xue et al., 2024). This reflects what Ruder et al. (2022) termed *Square One Bias*: progress in research often advances along a single axis (e.g., English alignment or multilingual coverage), while the intersection—multilingual calibration—remains underexplored. Existing studies examine few languages, rely on machine-translated data, and consider a narrow set of models (Xue et al., 2024; Yang et al., 2023).

To address this gap, we conduct the first large-scale systematic studies of multilingual calibration. Our analysis spans six major model families and over 100 languages, using high-quality, human-curated benchmarks (MMMLU and Belebele) with $\sim 10^5$ instances. Our findings reveal a stark disparity. Non-English languages consistently exhibit not only lower accuracy but also dramatically worse calibration—for instance, LLaMA-3's Expected Calibration Error (ECE) is nearly five times higher on average for non-English than for English (23.1% vs. 4.6%). Prior work often treats miscalibration in LLMs primarily as overconfidence (Zhang et al., 2024d; Chhikara, 2025). We instead reveal **distinct miscalibration patterns** tied to training priorities. English-aligned models like LLaMA3 struggle to maintain calibration quality beyond English, whereas multilingual-first models like Aya

---

[1]In this paper, we distinguish between calibration as a *property* and as a *process*. We use the term *calibration* to refer to the property of a model's confidence being well-aligned with its accuracy. In contrast, the methods used to achieve this alignment are referred to as *calibration methods* or a *calibrator*.

are systematically over-confident across the board. This suggests that current alignment strategies fail to generalize, creating unreliable models for non-English languages.

Seeking to understand the architectural source of this miscalibration, we challenge the conventional practice of extracting confidence scores solely from the model's final layer. Inspired by work showing intermediate layers encode more language-agnostic representations (Bandarkar et al., 2024b; Wendler et al., 2024), we hypothesize that the final layer, heavily **biased** by English-dominated training, provides a **sub-optimal** signal for multilingual confidence. A comprehensive layer-wise analysis reveals an interesting dichotomy: while English calibration improves *monotonically* with model depth, peaking at the final layer, multilingual calibration follows *a different trajectory*. For nearly all non-English languages, we find that *late-intermediate layers* consistently provide better-calibrated confidence estimates. This discovery of a latent, more reliable calibration signal hidden deeper within the model's architecture is a key finding of our work.

This core insight motivates our primary methodological contribution: a set of simple yet effective training-free calibration methods that leverage these intermediate representations. We compare three confidence elicitation strategies: the *best layer* method identifies and selects the single most calibrated intermediate layer, the *good layers-ensemble* approach aggregates signals from multiple layers to improve robustness, and finally we propose *LACE* (Language-aware Confidence Ensemble), a novel method that adaptively tailors layer selection to each target language. Our methods yield substantial and consistent improvements in multilingual calibration across all models. Crucially, they are orthogonal and complementary to traditional post-hoc techniques; combining *LACE* with calibration methods like Temperature Scaling (Guo et al., 2017) leads to further improvements.

Our contributions are threefold: (1) We provide a comprehensive empirical analysis of multilingual LLM calibration, revealing systematic and significant disparities between English and over 100 other languages. (2) We are the first to conduct a layer-wise investigation of multilingual calibration, discovering that intermediate layers offer a more reliable calibration signal than the final, English-biased layer. (3) We introduce novel, training-free calibration methods that leverage intermediate representations, demonstrating their effectiveness in closing the cross-lingual calibration gap.

## 2 RELATED WORK

**Multilingual Calibration**    Recent work has highlighted that modern LLMs, despite their strong performance, frequently struggle with calibration in their predictions (Xiong et al., 2024; Zhang et al., 2024c). Parallel studies document language-specific biases in LLMs (Zhang et al., 2024a; Qin et al., 2025). Yet calibration in multilingual settings remains underexplored. Ahuja et al. (2022) first established that multilingual models like mBERT and XLM-R are poorly calibrated, especially for low-resource languages like Swahili. Xue et al. (2024) conducted a confidence estimation study across various models, covering both language-agnostic and language-specific tasks, but datasets in their study included only 5 languages and were machine-translated which can potentially import bias (Vanmassenhove et al., 2021; Choenni et al., 2024). Our work distinguishes itself by presenting the first systematic evaluation of multilingual calibration across high-quality, human-curated datasets spanning over 100 languages and covering six prominent LLM families. Additionally, all prior studies have primarily documented calibration issues at the final output layer, none have examined confidence behaviour in depth or investigated its architectural origins, leaving the gap for our research.

**Layer-wise Representations**    A growing body of research investigates the functional specialization of layers within multilingual transformers. It is widely observed that intermediate layers encode cross-lingual semantic knowledge in a largely language-agnostic manner, forming a shared representational space (Bandarkar et al., 2024b). In contrast, the final layers tend to be more language-specific, adapting these general representations to handle surface-level features like syntax and word order for the target language. Recent studies on predominantly English-trained LLMs, such as LLaMA, suggest a more specific mechanism: the middle layers tend to operate in a largely language-agnostic space, where multilingual inputs are mapped into shared internal representations that often resemble English-like structures (Wendler et al., 2024; Kojima et al., 2024; Alabi et al., 2024). This hypothesis highlights that while surface forms differ across languages, the model internally normalizes them into a common representational layer before decoding back into the target language in the final layers, which explains the empirical success of prompting strategies that explicitly ask the model to

| Language | LLaMA3 | | | | Aya | | | |
|---|---|---|---|---|---|---|---|---|
| | ECE↓ | BRIER↓ | AUROC↑ | Accuracy | ECE↓ | BRIER↓ | AUROC↑ | Accuracy |
| Arabic | 33.06 | 24.37 | 61.00 | 38.20 | 28.41 | 33.79 | 71.49 | 45.20 |
| Bengali | 24.93 | 23.39 | 58.44 | 35.20 | 29.01 | 31.48 | 60.01 | 31.30 |
| German | 25.81 | 24.92 | 65.36 | 44.40 | 26.54 | 33.51 | 69.70 | 53.00 |
| Spanish | 18.21 | 21.89 | 71.65 | 52.00 | 28.17 | 31.86 | 71.12 | 51.10 |
| French | 13.87 | 22.75 | 71.39 | 51.30 | 23.80 | 32.72 | 70.69 | 53.40 |
| Hindi | 28.31 | 24.28 | 62.07 | 39.90 | 30.21 | 34.98 | 70.08 | 42.30 |
| Indonesian | 19.67 | 23.76 | 66.25 | 45.00 | 27.88 | 31.54 | 70.85 | 51.20 |
| Italian | 21.19 | 22.74 | 71.57 | 51.80 | 26.65 | 30.33 | 71.76 | 52.70 |
| Japanese | 28.36 | 27.27 | 61.73 | 43.00 | 16.30 | 26.26 | 69.92 | 46.70 |
| Korean | 30.86 | 25.06 | 62.59 | 42.50 | 32.07 | 37.09 | 72.06 | 45.00 |
| Portuguese | 10.51 | 21.76 | 71.37 | 50.40 | 27.33 | 31.42 | 70.71 | 53.50 |
| Swahili | 23.84 | 21.45 | 61.10 | 32.20 | 32.01 | 36.72 | 58.23 | 31.30 |
| Yoruba | 8.18 | 19.43 | 58.00 | 27.40 | 30.11 | 28.56 | 60.73 | 26.40 |
| Chinese | 41.94 | 19.56 | 50.63 | 23.10 | 17.12 | 28.75 | 67.35 | 52.20 |
| English | **4.61** | **17.63** | **80.36** | **61.20** | 20.66 | 25.30 | 74.65 | **57.40** |
| *Avg. Non-English* | 23.12 | 22.95 | 64.06 | 41.47 | 26.77 | 31.97 | 68.26 | 45.49 |
| *Avg. Low-Resource* | 23.00 | 22.78 | 61.14 | 36.32 | 29.60 | 32.84 | 65.23 | 37.95 |
| *Avg. High-Resource* | 21.71 | 22.62 | 67.41 | 46.63 | 24.29 | 30.80 | 70.88 | 51.67 |
| *Avg. Non-Latin-Script* | 27.44 | 23.10 | 59.44 | 35.19 | 26.90 | 32.20 | 66.23 | 40.05 |
| *Avg. Latin-Script* | 16.27 | 22.21 | 71.14 | 50.87 | 25.86 | 30.95 | 71.35 | 53.19 |
| *Average (All)* | 22.22 | 22.68 | 64.90 | 42.51 | 26.42 | 31.62 | 68.62 | 46.18 |

Table 1: Multilingual performance of **LLaMA3** (left) and **Aya** (right) on the MMMLU dataset. Metrics include ECE, Brier Score, AUROC, and Accuracy. All numbers are in percentages.

"think in English" before generating a response in another language, as this aligns with the model's internal processing pathway (Shi et al., 2023; Zhang et al., 2024e). Our work builds on these insights by investigating how this layer-wise specialization—particularly the language-neutral properties of intermediate representations—affects calibration across languages.

## 3 BENCHMARKING MULTILINGUAL CALIBRATION

In this section, we systematically examine the multilingual calibration in leading LLMs. We first detail our experimental setup using human-curated datasets, and then present our analysis of the models' performance across a diverse set of over 100 languages.

### 3.1 EXPERIMENTAL SETUP

**Datasets and Models** We focus on multilingual Multiple-Choice Question Answering (MCQA) because it offers precise control over both correctness and confidence. First, MCQA provides unambiguous correctness labels for calibration. In open-ended generation, correctness is typically approximated using metrics such as ROUGE, BLEU, or LLM-as-a-judge, which often disagree and can even lead to flipped conclusions depending on the choice of metric (Ielanskyi et al., 2025). Second, MCQA enables a clear definition of confidence as the probability assigned to the selected option. In contrast, confidence estimation for open-ended generation remains unstable and method-dependent (e.g., verbalized confidence, perplexity-based methods, or sampling-based consistency). By using MCQA, we can isolate the effect of layer-wise signals on calibration without additional noise from generation-based confidence heuristics.

We use the following two datasets: (1) **MMMLU** (Hendrycks et al., 2021) (15 languages), (2) **Belebele** (Bandarkar et al., 2024a) (122 languages). Compared to previous works, the datasets we use consist of high-quality human-translated items, covering a much larger range of languages and much more data points (at the scale of $\sim 10^5$ instances). All experiments are conducted using an eight-shot prompting setup in its respective language. We evaluate a range of recent LLMs, including **LLaMA3**, **Qwen2.5**, **Mistral**, **Aya**, **DeepSeek**, and **Phi** (see Appendix B.1 for details).

**Correctness Calculation and Confidence Elicitation** We take the output of the final layer as the model answer when evaluating correctness. We do not compute calibration using each layer's own prediction, because our goal is to provide a reliability signal for the deployed model in a

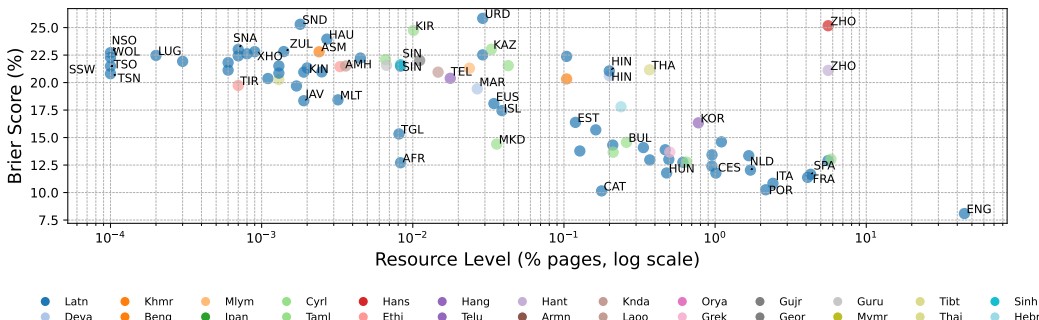

Figure 1: Relationship between resource level and Brier score for the LLaMA3 model on the Belebele benchmark. Each point represents a language, and same colour indicates same writing system. Correlations: Spearman $\rho = -0.59$, $p < 10^{-8}$; Kendall $\tau = -0.43$, $p < 10^{-8}$; Pearson $r = -0.39$, $p < 0.001$; indicating that higher-resourced languages tend to achieve better calibration.

standard inference setting, where the user only observes the final-layer answer. Therefore, the confidence score should approximate *the probability that this final-layer answer is correct*. We follow the standard MCQA practice and use the output probability of the selected answer choice: $\text{Conf}(x) = \max_{i \in \{1,\ldots,C\}} p_i$, where $\sum_{i=1}^{C} p_i = 1$, where $C$ is the number of choices.

**Metrics** We evaluate calibration using expected calibration error (ECE; Guo et al., 2017) and the Brier score (Brier, 1950). To measure model's ability to discriminate between correct and incorrect predictions, we also report AUROC (Fawcett, 2006). Lower ECE and Brier scores indicate better calibration; higher AUROC indicates stronger discrimination ability. We report ECE with number of bins $K = 10$, but to assess robustness to the number of bins, we additionally conduct experiments on binning sensitivity and observe a consistent discovery across different settings (see Appendix E.2).

## 3.2 RESULTS

Our results of models (LLaMA3 and Aya), shown in Table 1 for MMMLU and visualized in Figure 1 for Belebele (see Appendix B.3 for group definitions). Additional MMMLU results are provided in the Appendix, including Mistral (Table 3), Qwen2.5 (Table 4), Phi (Table 5), and Deepseek (Table 6). Comprehensive Belebele results for all models appear in Table 8, 9, 10, 11, 12, and 13 (see Appendix B.5). Our key findings are as follows:

**Not only are LMs more accurate but also more calibrated in English.** As shown in Table 1, non-English languages consistently underperform English in both accuracy and calibration. The average ECE for non-English LLaMA3 is 23.12%, far higher than the 4.61% for English, and Aya shows a similar pattern (26.77% v.s. 20.66%), highlighting that the language imbalance persists despite claims of improved multilingual capabilities (Dang et al., 2024). This discrepancy is evident in Brier Score and also AUROC. We also observe that Non-English languages show a much higher proportion of *underconfident correct predictions*—where model predicts correctly but has less than 50% confidence—at 78.8% compared to only 25.7% for English (see Table 7 in Appendix B.4). Moreover, in English the model assigns on average 23.8% higher confidence to predictions that are correct than to incorrect, whereas in non-English languages this margin is only 6.3%.

**Calibration correlates with language resource availability.** Table 1 suggests a calibration gap between low-resourced and high-resourced languages, for example, Hindi and Swahili show a comparatively worse calibration in both models. To further illustrate this, we plot the resource level[2] and calibration for all languages in Figure 1. We find that low-resource languages generally have much higher calibration error. We observe Spearman's correlation $\rho = -0.59$ ($p < 10^{-9}$) with

---

[2]Since LLaMA3 does not release the exact training data, we use the Common Crawl dataset (`CC-MAIN-2025-30`; Common Crawl Foundation, 2025)'s percentage of web pages available per language from the crawl as a proxy for global resource availability across languages.

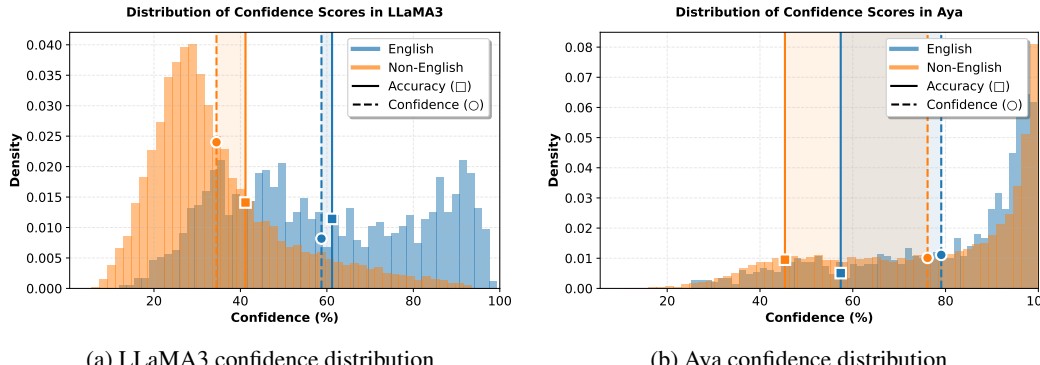

(a) LLaMA3 confidence distribution  (b) Aya confidence distribution

Figure 2: Confidence distributions for English v.s. Non-English samples in (a) LLaMA3 and (b) Aya models. The histograms show the density of model confidence scores. The overall distributions differ substantially between English and Non-English inputs in LLaMA3, and the gap between confidence (dashed lines) and accuracy (solid lines) is much larger for Aya.

Brier score and $\rho = 0.66$ ($p < 10^{-12}$) with AUROC, indicating that data-rich languages are better calibrated and show stronger discrimination ability. This pattern suggests that calibration is influenced by the representation of a language in the pre-training corpus.

***Square One Bias*: Differences in confidence distribution reflect training/alignment priorities.** Plotting the confidence distributions of the tested models reveals distinct calibration patterns that reflect their training priorities. For LLaMA-3 (Figure 2a), the English setting shows good calibration (ECE = 4.61), whereas the non-English setting exhibits a different confidence curve and greater under-confidence. By contrast, Aya's confidence distributions in English vs.non-English are similar in shape (Figure2b) but are strongly miscalibrated in both, showing a significant right skew (overconfidence). We argue that these patterns reflect the models' training and alignment policies. LLaMA-3 is documented for alignment efforts through supervised fine-tuning, preference ranking, and safety pipelines (Grattafiori et al., 2024), but this work appears primarily English-focused, leaving calibration in other languages largely unaddressed. Aya, by contrast, prioritizes multilingual coverage (Üstün et al., 2024; Dang et al., 2024) but pays less attention to calibration or caution in predictions. Together, these results echo the *Square One Bias* (Ruder et al., 2022): LLaMA-3 advances mainly in English alignment and safety, while Aya advances in multilingual ability, each neglecting the complementary dimension required for robust multilingual calibration. Confidence behaviour of other models can be found in Figure 8 in Appendix B.4.

## 4 MID-LAYERS REVEAL BETTER CALIBRATION

Inspired by recent insights in layer-wise multilingual representations (Bandarkar et al., 2024b; Wendler et al., 2024), we examine how confidence evolves throughout the model's depth to understand the source of the poor calibration observed in §3. Our analysis reveals that the final layers, which are over-specialized in English, can **harm the calibration for other languages**.

### 4.1 METHODOLOGY FOR EARLY-DECODED CONFIDENCE ESTIMATION

We investigate the following question: *Is it possible to identify intermediate representations that elicit better calibrated confidence scores with respect to final-layer accuracy?* We adopt a layer-wise probing technique inspired by the early exiting paradigm (Elbayad et al., 2020) to offer a new way of confidence estimation. Instead of applying the modelling head only to the final hidden state, we attach it to each intermediate transformer layer. This allows us to extract logits and compute prediction confidence from every layer, providing a granular view of the model's decision-making process.

Formally, let $\mathbf{h}_\ell \in \mathbb{R}^d$ denote the hidden representation at layer $\ell$, where $\ell = 1, \ldots, L$, and $d$ is the dimensionality of the hidden state. We apply the original language modeling head, with weight

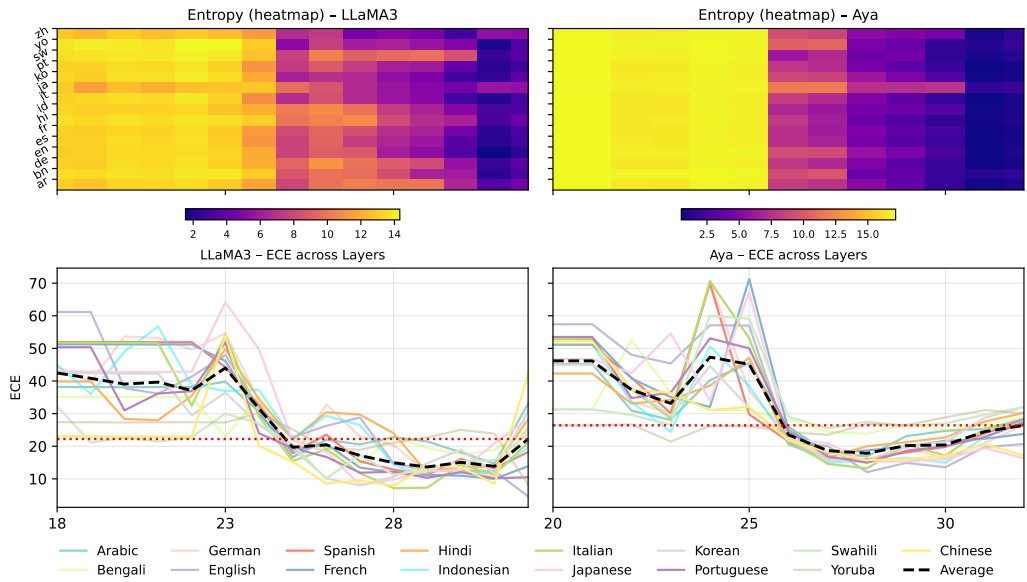

Figure 3: ECE v.s. entropy across layers on the MMMLU subset for LLaMA3 and Aya. In the multilingual setting, many languages achieve their **best** ECE in **intermediate layers** (e.g., 25-32 for LLaMA3 and 26-32 for Aya), after which calibration quality degrades towards the final layer. This contrasts with the English-only setting, where calibration improves monotonically (see Figure 9). Notably, the *sweet spot* in calibration coincides with the sharp drop in entropy.

matrix $W \in \mathbb{R}^{V \times d}$, to compute the logits at each layer:

$$\mathbf{z}_\ell = W \mathbf{h}_\ell$$

where $\mathbf{z}_\ell \in \mathbb{R}^V$ are the unnormalized token logits over the vocabulary of size $V$. These logits are then converted into probabilities using the softmax function:

$$\mathbf{p}_\ell = \text{softmax}(\mathbf{z}_\ell), \quad \sum_{v=1}^{V} [\mathbf{p}_\ell]_v = 1,$$

With the $\mathbf{p}_\ell$, we *trace* the token ultimately predicted at the final layer, $\hat{y}_L = \arg\max_v [\mathbf{p}_L]_v$, back through the intermediate layers. At each layer $\ell$, we then define the confidence score as the probability mass that this layer assigns to the final prediction, calibration is then evaluated by comparing these $\text{Conf}_\ell(x)$ with the prediction accuracy determined at the final layer $\hat{y}_L$:

$$\text{Conf}_\ell(x) = [\mathbf{p}_\ell]_{\hat{y}_L}, \quad \text{ECE}_\ell = \text{ECE}\Big(\{(\text{Conf}_\ell(x), \ \mathbf{1}\{\hat{y}_L = y\})\}\Big).$$

where $\mathbf{1}\{\hat{y}_L = y\}$ is the indicator function of whether the final-layer prediction is correct.

### 4.2 MULTILINGUAL LANGUAGE MODELS CALIBRATE EARLIER

**Last layer shows best calibration level in the English-only setting.** As shown in Figure 9 (see Appendix C.1) for LLaMA3, our layer-wise analysis shows a clear trend: the layer-predicted confidence calibration improves monotonically with layer depth. This aligns with the conventional understanding (Tenney et al., 2019) that representations become progressively more refined and task-specific, leading to better calibration as data propagates through the network.

**Multilingual settings reveal best calibration in late intermediate layers.** However, our analysis reveals a different pattern in the multilingual context. As illustrated in Figure 3, the best calibration performance for many languages **does not** occur at the final layer. Instead, we find a *sweet spot* with lower ECE in the late-intermediate layers (between layers 24-end for LLaMA3 and 26-end for Aya, both are 32-layer models), after which calibration quality *worsens* to the final output layer.

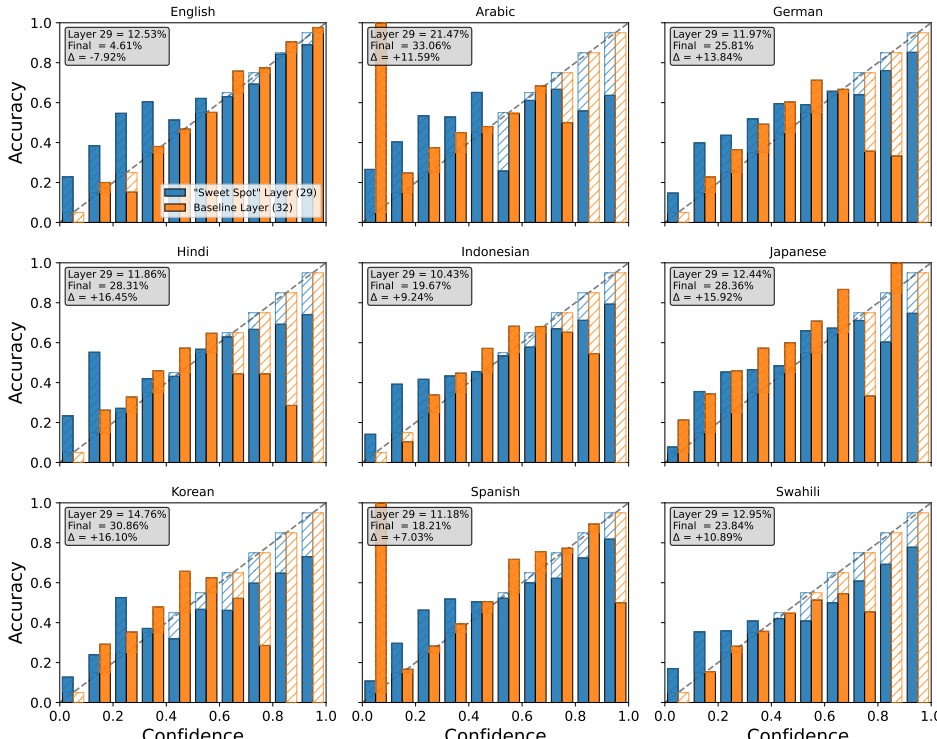

Figure 4: **Per-language calibration reliability diagrams for LLaMA3.** Each panel shows a reliability histogram with evenly spaced confidence bins. **Blue** bars correspond to the chosen intermediate layer (Layer 29), and **orange** bars correspond to the original final layer. The dashed diagonal is the perfectly calibrated line ($y = x$). Hatched overlays indicate the absolute calibration gap within each bin. The inset reports ECE (%) for both layers and the change $\Delta\text{ECE} = \text{ECE}_{\text{Final}} - \text{ECE}_{29}$ (positive values denote improved calibration at Layer 29).

Concretely, for LLaMA3, selecting layer 29 results in an average $\Delta$ECE of 8.57, while for Aya layer 28 results a comparable 8.59. Notably, this turning point in calibration quality aligns with the trend in entropy: as entropy begins to drop sharply in these intermediate layers, ECE also decreases.

**Per-language calibration trends reveal that late-intermediate layer improves calibration for most non-English languages, with a slight trade-off for English.** To further explore these dynamics, Figure 16 presents per-language reliability diagrams for nine languages, comparing the selected intermediate layer (Layer 29) against the final layer of LLaMA3. Nearly all non-English languages benefit greatly from moving away from the final layer: their reliability curves align more closely with the diagonal, and ECE decreases substantially. For example, German and Hindi show reductions in ECE of more than 13% and 16%, respectively. In contrast, calibration for English appears to degrade slightly at intermediate layers. Since the final layer already exhibits strong calibration for English, earlier layers offer no additional benefit. This highlights a potential bias introduced during pretraining, where the model overfits to English patterns or introduces noise during the final stages of adaptation. While the degree of improvement varies across languages, the trend remains consistent across diverse linguistic families and scripts. This suggests that the effect is not language-specific but rather a systematic property of multilingual calibration. Additional examples are provided in Appendix C.3.

**The mid-layer calibration peak is a robust finding across models.** Our observation is not isolated to a single model or metric. We consistently find this pattern across multiple architectures and evaluation metrics, as detailed in the Appendix C.2. In LLaMA-3.1 8B (Figure 10), Aya (Figure 11), Mistral (Figure 12), and others, multilingual models exhibit a latent calibration optimum not at the decoding layer but in late-intermediate layers. To demonstrate that this finding generalizes beyond smaller models and remains present in substantially larger architectures, we provide additional

evidence for LLaMA-3.1 70B (Figure 18, 19, 20). This finding challenges the common practice to use final-layer probabilities to calculate model confidence, and it opens avenues for layer-aware calibration strategies that explicitly exploit these "sweet spots" to mitigate cross-lingual disparities, which motivates the novel calibration methods proposed in the next section.

## 5 IMPROVING MULTILINGUAL CALIBRATION

### 5.1 MULTILINGUAL CALIBRATION METHODS

Our observations in Section 4 motivate confidence estimators that exploit intermediate representations. We explore to extract confidence in three different ways:

- **Final layer (baseline).** We follow the standard practice in prior work by using the model's final layer to derive probabilities, which serves as our baseline.
- **Best layer.** We identify the *best layer* as the one that achieves the lowest average calibration error across languages. From this layer, we extract probability estimates following the procedure in Section 4.1. The best layer $\ell^*$ is selected using a separate validation set and defined as:

$$\ell^* \; = \; \arg \min_{\ell \in \{1,\ldots,L\}} \mathrm{ECE}_\ell^{\mathrm{avg}}.$$

- **Good layers ensemble.** We consider the set of layers (*good layers*) whose multilingual calibration outperforms the final layer. To obtain confidence estimates, we average the predictive distributions across these layers $\mathcal{G}$, this reduces layer-specific noise while pooling calibration-aware signals:

$$\mathcal{G} \; = \; \left\{\ell : \mathrm{ECE}_\ell^{\mathrm{avg}} < \mathrm{ECE}_L^{\mathrm{avg}}\right\}, \qquad \mathbf{p}_{\mathrm{ensemble}} \; = \; \frac{1}{|\mathcal{G}|} \sum_{\ell \in \mathcal{G}} \mathbf{p}_\ell.$$

**Combining with Classical Post-hoc Calibration.** Since confidence elicitation and calibration are orthogonal approaches, we further test whether the proposed elicitation methods can be enhanced by standard post-hoc calibration techniques such as Temperature Scaling (Guo et al., 2017) and Isotonic Regression (Zadrozny & Elkan, 2002), which operate independently of how probabilities were obtained (Kadavath et al., 2022; Minderer et al., 2021). We adopt a two-stage pipeline:

$$\mathbf{p}_{\mathrm{final}} \; = \; \mathrm{Calibrate}(\mathbf{p}_{\mathrm{intermediate}}),$$

where $\mathbf{p}_{\mathrm{intermediate}}$ comes from the confidence scores discussed above. See Appendix D for details.

**Language-Aware Confidence Ensemble.** The approaches described above work in a global setting that optimizes for holistic performance across languages. However, if we aim to optimize for a specific language, we can pursue more tailored strategies to address unique calibration dynamics. To that end, we introduce a novel approach *Language-Aware Confidence Ensemble* (LACE), inspired by our layerwise analysis (Section 4) and by language-specific methods that adaptively use different layers for different languages.

For each language $k$, we predict confidence from layers that are better calibrated than the final layer,

$$\mathcal{G}^{(k)} \; = \; \{\ell \; : \; \mathrm{ECE}_\ell^{(k)} < \mathrm{ECE}_L^{(k)}\}, \quad \mathbf{p}_{\mathrm{ensemble}}^{(k)} \; = \; \tfrac{1}{|\mathcal{G}^{(k)}|} \sum_{\ell \in \mathcal{G}^{(k)}} \mathbf{p}_\ell^{(k)},$$

and learn a language-specific calibrator mapping:

$$\mathbf{p}_{\mathrm{final}}^{(k)} \; = \; \mathrm{Calibrate}^{(k)}\big(\mathbf{p}_{\mathrm{ensemble}}^{(k)}\big).$$

Theoretically, LACE is effective for two reasons. **First**, per-language layer selection avoids negative transfer from layers that are miscalibrated for the target language. **Second**, ensembling over the selected layers reduces variance while preserving language-relevant signals.

Meanwhile, LACE has two main advantages. **First**, LACE is modular and training-free. Retraining or fine-tuning LLMs is computationally prohibitive for most practitioners. LACE provides a training-free, plug-and-play solution that can be directly applied to deployed models to improve safety without changing model weights. **Second**, LACE has low overhead: by reusing intermediate logits from the single forward pass already required for generation, it introduces negligible latency compared to computationally expensive sampling-based consistency checks.

| Method | MMMLU | | | | | | Belebele | | | | | |
| | LLaMA3 (Acc. = 43.2%) | | | Aya (Acc. = 48.8%) | | | LLaMA3 (Acc. = 68.6%) | | | Aya (Acc. = 67.8%) | | |
| | ECE | Brier | AUROC | ECE | Brier | AUROC | ECE | Brier | AUROC | ECE | Brier | AUROC |
|---|---|---|---|---|---|---|---|---|---|---|---|---|
| FINAL LAYER BASELINE[†] | 22.44 | 23.03 | 64.05 | 24.39 | 30.45 | 68.31 | 17.68 | 20.38 | 69.01 | 15.73 | 19.08 | **72.92** |
| ▷ (+Temperature Scaling) | 23.35 | 22.59 | 64.05 | 15.40 | 23.76 | 68.31 | 17.63 | 20.39 | 69.01 | 10.66 | 17.19 | 72.92 |
| ▷ (+Isotonic Regression) | 20.23 | 22.55 | 63.47 | 9.15 | 22.02 | 68.07 | 11.09 | 19.07 | 68.85 | 8.33 | 16.44 | 72.78 |
| *Intermediate Layer Calibration (Global Selection)* | | | | | | | | | | | | |
| BEST LAYER[†] | 14.28 | 22.78 | 71.44 | 17.57 | 27.08 | 66.68 | 13.67 | 18.28 | 71.33 | 15.40 | 20.06 | 66.79 |
| ▷ (+Temperature Scaling) | 13.71 | 20.60 | 71.44 | 9.34 | 22.52 | 66.68 | 13.12 | 17.52 | 71.34 | 14.99 | 19.16 | 66.79 |
| ▷ (+Isotonic Regression) | 13.12 | 20.80 | 71.28 | 10.66 | 22.46 | 66.26 | 12.40 | 17.39 | 71.48 | 14.02 | 19.00 | 66.80 |
| GOOD LAYERS ENSEMBLE[†] | 11.84 | 20.23 | **73.91** | 13.10 | 23.78 | **68.62** | 10.78 | 15.59 | **76.26** | 11.47 | 18.00 | 70.24 |
| ▷ (+Temperature Scaling) | 11.30 | 20.01 | 73.91 | 10.23 | 22.21 | 68.62 | 10.49 | 15.62 | 76.26 | 11.81 | 17.57 | 70.24 |
| ▷ (+Isotonic Regression) | 9.60 | 19.90 | 73.49 | 7.71 | 21.82 | 68.25 | 10.16 | 15.54 | 76.07 | 10.42 | 17.48 | 70.31 |
| *Intermediate Layer Calibration (LACE)* | | | | | | | | | | | | |
| LANGUAGE-AWARE ENSEMBLE[‡] | 5.96 | 20.51 | 72.94 | 11.42 | 22.70 | 68.38 | 7.05 | **14.35** | 75.61 | 10.22 | 17.77 | 69.79 |
| ▷ (+Temperature Scaling) | 4.34 | **19.73** | 73.40 | 4.88 | 21.87 | 68.49 | 6.05 | 14.47 | 74.98 | 5.46 | **16.36** | 70.45 |
| ▷ (+Isotonic Regression) | **3.09** | 20.51 | 69.13 | **3.45** | **21.46** | 67.10 | **5.79** | 14.53 | 73.70 | **4.80** | 16.73 | 68.40 |

Table 2: Calibration results for **LLaMA3** and **Aya** on MMMLU (left) and the Belebele subset (right). [†] denote methods that do not assume access to language identity. [‡] denote methods with given language identity. Indented italic rows correspond to post-hoc calibration adjustments.

**Experiment Setup**  We use the data from Section 4 as a held-out validation set and evaluate on a separate MMMLU test split. Both splits are balanced across languages, with a total of 30K examples. For Belebele, we construct a comparable evaluation set with 24K examples overall with a similar validation/test split. We report macro-averaged results across languages.

## 5.2 CALIBRATION RESULTS

**Intermediate-layer confidence shows better calibration than final-layer confidence.** Figure 5 shows that moving from Best Layer to Good-Layers Ensemble to LACE yields progressive improvements in calibration. Detailed numbers are reported in Table 2 for both MMMLU and Belebele. Note that occasional drops in AUROC occur because discrimination and calibration are not necessarily correlated (Gao et al., 2022; Carriero et al., 2024), and our layer selection is based on ECE rather than AUROC. Finally, by tailoring layer selection and calibration to each language, LACE achieves the best overall results, with ECE as low as 5.96 on LLaMA3.

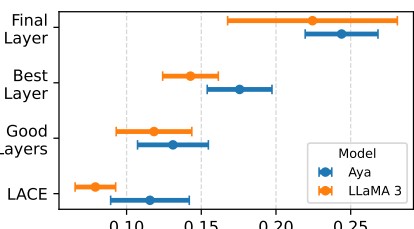

Figure 5: Forest plot of average ECE in MMMLU, with means and 95% CIs.

**Classical post-hoc methods offer improvements but face challenges in multilingual settings.** Temperature scaling and isotonic regression noticeably reduce the calibration error for Aya (systematically overconfident, see Figure 2; Aya has 76.9% average confidence on MMMLU and 82.5% on Belebele). In this setting, flattening the confidence curve provides clear gains. However, these methods offer marginal benefit or even degradation for LLaMA3 (Avg. Conf. = 36.0% for MMMLU and 58.3% for Belebele), suggesting a global temperature fitted on the multilingual validation set might limited little benefit for individual languages due to distributional heterogeneity (see Simpson (1951)'s paradox). More broadly, while such methods adjust prediction confidence through global rescaling, they often struggle to deliver consistent improvements across languages and model families. This calls for finer-grained strategies or more structurally integrated calibration methods (Hébert-Johnson et al., 2018).

**LACE is complementary to post-hoc approaches and delivers the best calibration performance.** Importantly, our method *reshapes*, rather than *rescales*, the calibration signal (e.g., LLaMA3 baseline Avg. Conf=35.97% v.s. LACE Avg. Conf=36.57%). Therefore, our method does not compete

with post-hoc calibrators but provides orthogonal improvements. As shown in Table 2, all methods combined with temperature scaling or isotonic regression yields further incremental gains, and notably LACE is further boosted to consistently deliver the lowest calibration error across benchmarks: ECE to 3.09 on LLaMA3 and 3.45 on Aya. This complementarity highlights the practical value of our approach as a flexible, additive pathway toward reliable multilingual calibration.

## 5.3 ANALYSIS

| Method | ECE | Brier | AUROC |
|---|---|---|---|
| ORIG. PROB. | 23.51 | 22.10 | 65.19 |
| (+Temp. Scal.) | 22.56 | 21.97 | 65.19 |
| (+Isotonic) | 19.01 | 21.34 | 65.07 |
| LACE | 6.14 | 15.76 | 79.42 |
| (+Temp. Scal.) | 4.93 | 15.60 | 79.92 |
| (+Isotonic) | 2.81 | 15.38 | 78.79 |

Figure 6: Calibration for LLaMA-3.1 70B on MMLU.

**LACE scales effectively to larger models.** To show that our approach is not limited to medium-sized models, we also include experimental results for larger models (Table 6; full results in Appendix E.1). Consistent with our findings at 7B models, our layer-based methods continue to outperform final-layer confidence in 70B models. LACE provides the largest overall improvement at this scale, reducing ECE from 23.51 (baseline) to as low as 2.81 when combined with Isotonic Regression.

**Layer-set overlap across languages and datasets.** To verify that LACE captures robust calibration signals that generalize beyond specific samples or datasets, we examine the consistency of learned layer sets across different languages and benchmarks. We find that *similar languages tend to share overlapping layer subsets*. As shown in Figure 7, we see clusters for high overlaps between Western European languages and CJK (Chinese, Japanese, Korean) languages (highlighted in red box). This structured similarity suggests partially shared calibration-relevant signals across linguistic groups (full discussion in Appendix E.4). Our cross-dataset evaluation (Appendix E.5) also shows that good-layer sets derived from Belebele retain strong calibration performance when transferred directly to MMLU, outperforming the baseline without additional tuning. Collectively, the observed linguistic clustering and successful cross-task transfer confirm that LACE leverages stable, structurally consistent representations rather than overfitting to dataset-specific artifacts.

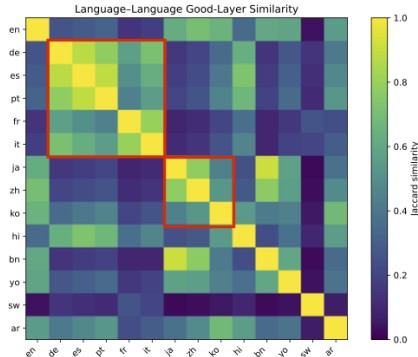

Figure 7: Jaccard Similarity between different language's layer-sets in LLaMA-3.1 70B.

**LACE is robust to moderate language-ID noise.** Building on the observation that languages share overlapping layer sets, we next examine whether this structural similarity translates into robustness when the language routing signal is unreliable, i.e., when the input language is uncertain. In our controlled corruption experiment (Appendix E.3) in which we randomly replace the language tag of test examples, LACE shows only minor degradation across all metrics. This indicates that the overlapping layer structure enables LACE to remain stable even under moderate language-ID noise.

## 6 CONCLUSION

In this work, we present the first systematic studies of multilingual calibration in large language models. Our findings highlight stark disparities between English and other languages: models exhibit not only lower accuracy outside English but also severe miscalibration. Our analysis reveals that calibration quality is not uniform across depth: while English benefits from final-layer confidence signals, multilingual reliability emerges more strongly in intermediate representations. Building on this insight, we propose a family of training-free calibration methods that leverage these intermediate layers. We introduce the adaptive *LACE* method and demonstrate consistent, substantial improvements in multilingual calibration. Moreover, we show that these methods complement traditional post-hoc techniques, enabling state-of-the-art calibration. We hope this work motivates future research at the intersection of multilinguality and calibration and ultimately contributes to more equitable and trustworthy deployment of language technologies worldwide.

ETHICS STATEMENT

Our research adheres to strict ethical guidelines. We verified the licenses of all software and datasets used in this study to ensure full compliance with their terms. No privacy concerns have been identified. We have conducted a thorough assessment of the project and do not anticipate any further risks. We only use LLMs for grammar checking during the paper writing.

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

## A LIMITATIONS

Our focus is on standard multiple-choice QA tasks as with such tasks model correctness is well-defined and easy to measure. The observed benefits of using intermediate layers may not directly extend to open-ended generative tasks such as dialogue, summarization, or long-form QA: we leave those tasks for future research. Also, our proposed methods are post-hoc interventions that correct poor calibration, rather than fundamental solutions that integrate multilingual calibration objectives into the model's training process to address the issue at its root. This constitutes another very compelling direction for future research.

## B BENCHMARKING MULTILINGUAL CALIBRATION

In this section, we present the detailed multilingual evaluation results for the models and benchmarks discussed in the main text.

### B.1 MODELS

Our experiments evaluate recent multilingual large language models:

- **LLaMA3** (Grattafiori et al., 2024) (If **not** specified: `Llama-3.1-8B-Instruct`)
- **Qwen2.5** (Yang et al., 2024a) (`Qwen2.5-7B-Instruct`)
- **Mistral** (Jiang et al., 2023) (`Mistral-7B-Instruct-v0.3`)
- **Aya** (Dang et al., 2024) (`aya-expanse-8b`)
- **DeepSeek** (DeepSeek-AI, 2025) (`DeepSeek-R1-Distill-Qwen-7B`)
- **Phi** (Abdin et al., 2024) (`phi-4`)
- **LLaMA-3.1 70B** (Grattafiori et al., 2024) (`Llama-3.1-70B-Instruct`)

### B.2 MMMLU RESULTS

| Language | AUROC | ECE | BRIER | Accuracy |
|---|---|---|---|---|
| Arabic | 64.91 | 41.18 | 11.87 | 4.50 |
| Bengali | 64.56 | 49.70 | 11.72 | 0.10 |
| German | 70.84 | 24.14 | 29.32 | 43.00 |
| English | 73.75 | 23.92 | 27.95 | 54.00 |
| Spanish | 71.33 | 21.64 | 26.79 | 42.90 |
| French | 71.25 | 22.20 | 28.36 | 46.40 |
| Hindi | 75.08 | 39.77 | 6.23 | 1.60 |
| Indonesian | 69.48 | 26.98 | 29.69 | 38.80 |
| Italian | 74.08 | 25.24 | 28.25 | 44.50 |
| Japanese | 56.09 | 44.15 | 15.48 | 6.50 |
| Korean | 39.78 | 46.62 | 16.25 | 5.50 |
| Portuguese | 71.11 | 29.25 | 27.59 | 47.10 |
| Swahili | 56.02 | 30.81 | 27.34 | 26.30 |
| Yoruba | 44.79 | 44.18 | 21.99 | 16.10 |
| Chinese | 62.12 | 33.55 | 24.58 | 16.70 |
| *Avg. Low-Resource* | 62.47 | 38.77 | 18.14 | 14.57 |
| *Avg. High-Resource* | 65.59 | 30.08 | 24.95 | 34.07 |
| *Avg. Latin-Script* | 71.69 | 24.77 | 28.28 | 45.24 |
| *Avg. Non-Latin-Script* | 57.92 | 41.24 | 16.93 | 9.66 |
| ***Average (All Languages)*** | **64.35** | **33.56** | **22.23** | **26.27** |

Table 3: Performance comparison across languages for AUROC, ECE, BRIER score, and Accuracy in **Mistral**, evaluated on the MMMLU dataset.

Results for each model are reported in the following tables: Mistral (Table 3), Qwen2.5 (Table 4), Phi (Table 5), and DeepSeek (Table 6).

| Language | AUROC | ECE | BRIER | Accuracy |
|---|---|---|---|---|
| Arabic | 67.15 | 14.30 | 26.67 | 54.90 |
| Bengali | 64.10 | 26.68 | 31.98 | 33.20 |
| German | 76.94 | 21.59 | 25.08 | 55.60 |
| English | 78.23 | 15.77 | 19.25 | 65.60 |
| Spanish | 76.95 | 19.26 | 23.98 | 61.10 |
| French | 75.65 | 16.92 | 22.88 | 62.20 |
| Hindi | 72.01 | 28.73 | 28.86 | 33.90 |
| Indonesian | 75.69 | 15.83 | 23.53 | 54.30 |
| Italian | 75.32 | 21.07 | 24.46 | 58.70 |
| Japanese | 80.03 | 6.71 | 17.10 | 33.10 |
| Korean | 74.15 | 17.60 | 25.75 | 52.20 |
| Portuguese | 75.85 | 18.86 | 23.61 | 58.40 |
| Swahili | 59.93 | 30.12 | 33.09 | 32.30 |
| Yoruba | 23.49 | 46.99 | 36.11 | 2.00 |
| Chinese | 85.31 | 12.47 | 17.42 | 47.00 |
| *Avg. Low-Resource* | 60.40 | 27.11 | 30.04 | 35.10 |
| *Avg. High-Resource* | 77.60 | 16.69 | 22.17 | 54.88 |
| *Avg. Latin-Script* | 76.38 | 18.47 | 23.26 | 59.41 |
| *Avg. Non-Latin-Script* | 65.77 | 22.95 | 27.12 | 36.08 |
| *Average (All Languages)* | **70.72** | **20.86** | **25.32** | **46.97** |

Table 4: Performance comparison across languages for AUROC, ECE, BRIER score, and Accuracy in **Qwen 2.5**, evaluated on the MMMLU dataset.

| Language | AUROC | ECE | BRIER | Accuracy |
|---|---|---|---|---|
| Arabic | 52.66 | 30.21 | 25.35 | 36.50 |
| Bengali | 52.62 | 34.13 | 24.73 | 27.20 |
| German | 63.47 | 22.86 | 22.86 | 65.60 |
| English | 71.13 | 20.48 | 17.92 | 73.10 |
| Spanish | 61.29 | 27.15 | 25.32 | 56.40 |
| French | 71.57 | 17.07 | 20.21 | 68.90 |
| Hindi | 37.74 | 46.43 | 26.16 | 15.70 |
| Indonesian | 42.89 | 32.36 | 30.63 | 30.70 |
| Italian | 72.25 | 10.51 | 19.13 | 67.50 |
| Japanese | 30.62 | 46.69 | 17.59 | 8.30 |
| Korean | 66.95 | 29.00 | 24.50 | 50.00 |
| Portuguese | 73.79 | 13.24 | 18.77 | 66.60 |
| Swahili | 64.42 | 16.18 | 23.61 | 40.50 |
| Yoruba | 53.76 | 20.83 | 21.01 | 27.60 |
| Chinese | 59.73 | 31.98 | 26.17 | 44.60 |
| *Avg. Low-Resource* | 50.68 | 30.02 | 25.25 | 29.70 |
| *Avg. High-Resource* | 63.42 | 24.33 | 21.39 | 55.67 |
| *Avg. Latin-Script* | 65.20 | 20.52 | 22.12 | 61.26 |
| *Avg. Non-Latin-Script* | 52.31 | 31.93 | 23.64 | 31.30 |
| *Average (All Languages)* | **58.33** | **26.61** | **22.93** | **45.28** |

Table 5: Performance comparison across languages for AUROC, ECE, BRIER score, and Accuracy in **Phi**, evaluated on the MMMLU dataset.

### B.3 MMMLU LANGUAGE GROUP DEFINITIONS

We randomly sampled 1,000 examples per language for MMMLU. We group languages in the MMLU dataset according to resource availability and script as follows:

**Low-Resource Languages**   Languages with relatively limited annotated data and pretrained model support: Arabic, Bengali, Swahili, Yoruba, Hindi, Indonesian.

| Language | AUROC | ECE | BRIER | Accuracy |
|---|---|---|---|---|
| Arabic | 55.33 | 32.74 | 21.54 | 26.40 |
| Bengali | 58.50 | 40.80 | 14.41 | 13.70 |
| German | 60.28 | 18.50 | 23.91 | 39.80 |
| English | 66.21 | 9.10 | 22.92 | 47.10 |
| Spanish | 62.24 | 12.47 | 23.51 | 40.80 |
| French | 62.93 | 10.84 | 23.12 | 41.40 |
| Hindi | 56.08 | 30.42 | 20.62 | 26.40 |
| Indonesian | 61.00 | 31.61 | 21.11 | 27.30 |
| Italian | 63.14 | 5.65 | 22.85 | 40.40 |
| Japanese | 55.56 | 18.05 | 23.14 | 32.10 |
| Korean | 21.56 | 49.09 | 18.66 | 1.10 |
| Portuguese | 62.37 | 16.78 | 23.26 | 39.10 |
| Swahili | 51.67 | 45.76 | 12.45 | 12.00 |
| Yoruba | 60.35 | 38.16 | 4.94 | 2.80 |
| Chinese | 69.00 | 16.13 | 23.97 | 43.10 |
| *Avg. Low-Resource* | 57.16 | 36.58 | 15.84 | 18.10 |
| *Avg. High-Resource* | 58.14 | 17.40 | 22.82 | 36.10 |
| *Avg. Latin-Script* | 62.60 | 14.99 | 22.95 | 39.41 |
| *Avg. Non-Latin-Script* | 53.51 | 33.89 | 17.47 | 19.70 |
| ***Average (All Languages)*** | **57.75** | **25.07** | **20.03** | **28.90** |

Table 6: Performance comparison across languages for AUROC, ECE, BRIER score, and Accuracy in **DeepSeek**, evaluated on the MMMLU dataset.

**High-Resource Languages**  Languages with substantial resources and strong support in major multilingual models: German, French, English, Spanish, Chinese, Italian, Japanese, Korean, Portuguese.

**Latin-Script Languages**  Languages primarily written using the Latin script: German, English, Spanish, French, Indonesian, Italian, Portuguese.

**Non-Latin-Script Languages**  Languages primarily written using non-Latin scripts (e.g., Arabic script, Devanagari, Hangul, Han characters): Arabic, Bengali, Hindi, Japanese, Korean, Swahili, Yoruba, Chinese.

### B.4 MODEL CONFIDENCE BEHAVIOURS

Figure 8 illustrates the distribution of confidence scores and accuracies across English and non-English settings for Qwen2.5, DeepSeek, Mistral, and Phi. Solid vertical lines indicate mean accuracies, while dashed vertical lines indicate mean confidences. The divergence between confidence and accuracy highlights calibration behaviour: underconfidence when the dashed line falls left of the solid line, and overconfidence when it falls to the right.

Table 7 reports detailed calibration and confidence statistics for LLaMA3. In addition to standard accuracy, we provide the model's average confidence, the confidence gap (accuracy minus confidence), the proportion of correct predictions made with low confidence ("Underconf"), and the mean confidence levels assigned to correct vs. incorrect predictions. We further include the difference between these two distributions ("Corr–Inc Gap"), which captures how well the model separates correct from incorrect responses. English shows a relatively small confidence gap (2.5%), with strong separation between correct and incorrect predictions (23.8% Corr–Inc Gap). In contrast, most non-English languages show lower accuracy, larger underconfidence rates, and much smaller separation between correct and incorrect predictions (average Corr–Inc Gap of 6.3%).

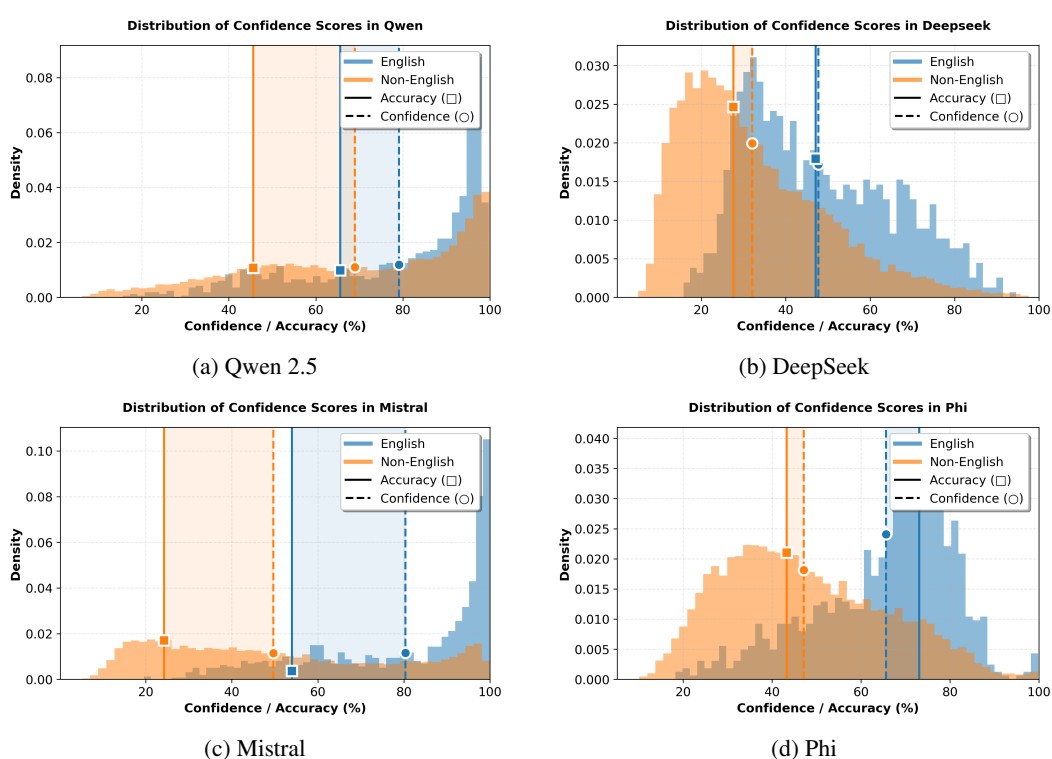

Figure 8: Distribution of confidence scores (predicted probabilities) versus accuracies in English (blue) and non-English (orange) for four models: (a) Qwen 2.5, (b) DeepSeek, (c) Mistral, and (d) Phi.

| Language | Acc. (%) | Avg Conf (%) | Conf Gap (%) | Underconf (%) | Corr. Conf (%) | Inc. Conf (%) | Corr–Inc Gap (%) |
|---|---|---|---|---|---|---|---|
| Arabic | 38.2 | 29.5 | 8.8 | 89.0 | 31.7 | 28.0 | 3.7 |
| Chinese | 23.1 | 23.2 | -0.1 | 99.6 | 22.2 | 23.6 | -1.4 |
| Korean | 42.5 | 31.2 | 11.3 | 92.0 | 33.5 | 29.5 | 4.0 |
| Japanese | 43.0 | 24.0 | 19.0 | 92.3 | 26.8 | 21.8 | 5.0 |
| Swahili | 32.2 | 31.5 | 0.7 | 88.8 | 34.2 | 30.3 | 3.9 |
| Italian | 51.8 | 41.8 | 10.0 | 58.3 | 47.4 | 35.8 | 11.6 |
| Bengali | 35.2 | 34.6 | 0.6 | 88.4 | 36.1 | 33.8 | 2.3 |
| Spanish | 52.0 | 46.1 | 5.9 | 50.0 | 52.6 | 39.0 | 13.6 |
| Portuguese | 50.4 | 47.1 | 3.3 | 47.0 | 53.6 | 40.6 | 13.0 |
| Indonesian | 45.0 | 37.0 | 8.0 | 75.3 | 41.1 | 33.7 | 7.4 |
| Hindi | 39.9 | 31.7 | 8.2 | 88.2 | 34.1 | 30.1 | 4.0 |
| German | 44.4 | 33.8 | 10.6 | 80.4 | 37.1 | 31.2 | 5.9 |
| Yoruba | 27.4 | 29.4 | -2.0 | 93.8 | 31.5 | 28.7 | 2.8 |
| French | 51.3 | 41.5 | 9.8 | 59.5 | 47.4 | 35.3 | 12.1 |
| **English** | 61.2 | 58.8 | 2.5 | **25.7** | 68.0 | 44.2 | **23.8** |
| **Non-English** | 41.2 | 34.5 | 6.7 | **78.8** | 37.8 | 31.5 | **6.3** |

Table 7: LLaMA3 Calibration and underconfidence analysis across languages. Metrics include accuracy, average confidence, confidence gap (accuracy minus confidence), proportion of underconfident correct predictions (confidence < 0.5 when correct), average confidence for correct vs. incorrect predictions, and their difference (Corr–Inc Gap).

## B.5    BELEBELE RESULTS

Belebele (Bandarkar et al., 2024a) is a multiple-choice dataset covering 122 language variants, enabling robust evaluation of NLU across high-, medium-, and low-resource languages. The dataset is fully parallel, allowing for direct cross-linguistic comparison of model performance. In our experiments, we sample 400 examples per language and evaluate the six model.

**Language Code**    The following FLORES-200 language codes (3-letter form) are included in the dataset evaluation:

| | | |
|---|---|---|
| acm - Mesopotamian Arabic | ilo - Ilocano | por - Portuguese |
| afr - Afrikaans | ind - Indonesian | ron - Romanian |
| als - Tosk Albanian | isl - Icelandic | rus - Russian |
| amh - Amharic | ita - Italian | shn - Shan |
| apc - North Levantine Arabic | jav - Javanese | sin - Sinhala |
| arb - Modern Standard Arabic | jpn - Japanese | slk - Slovak |
| ars - Najdi Arabic | kac - Jingpho | slv - Slovenian |
| ary - Moroccan Arabic | kan - Kannada | sna - Shona |
| arz - Egyptian Arabic | kat - Georgian | snd - Sindhi |
| asm - Assamese | kaz - Kazakh | som - Somali |
| azj - North Azerbaijani | kea - Kabuverdianu | sot - Southern Sotho |
| bam - Bambara | khk - Halh Mongolian | spa - Spanish |
| ben - Bengali | khm - Khmer | srp - Serbian |
| bod - Standard Tibetan | kin - Kinyarwanda | ssw - Swati |
| bul - Bulgarian | kir - Kyrgyz | sun - Sundanese |
| cat - Catalan | kor - Korean | swe - Swedish |
| ceb - Cebuano | lao - Lao | swh - Swahili |
| ces - Czech | lin - Lingala | tam - Tamil |
| ckb - Central Kurdish | lit - Lithuanian | tel - Telugu |
| dan - Danish | lug - Ganda | tgk - Tajik |
| deu - German | luo - Luo | tgl - Tagalog |
| ell - Greek | lvs - Standard Latvian | tha - Thai |
| eng - English | mal - Malayalam | tir - Tigrinya |
| est - Estonian | mar - Marathi | tsn - Tswana |
| eus - Basque | mkd - Macedonian | tso - Tsonga |
| fin - Finnish | mlt - Maltese | tur - Turkish |
| fra - French | mri - Maori | ukr - Ukrainian |
| fuv - Nigerian Fulfulde | mya - Burmese | urd - Urdu |
| gaz - West Central Oromo | nld - Dutch | uzn - Northern Uzbek |
| grn - Guarani | nob - Norwegian Bokmål | vie - Vietnamese |
| guj - Gujarati | npi - Nepali | war - Waray |
| hat - Haitian Creole | nso - Northern Sotho | wol - Wolof |
| hau - Hausa | nya - Nyanja | xho - Xhosa |
| heb - Hebrew | ory - Odia | yor - Yoruba |
| hin - Hindi | pan - Eastern Panjabi | zho - Chinese |
| hrv - Croatian | pbt - Southern Pashto | zsm - Standard Malay |
| hun - Hungarian | pes - Western Persian | zul - Zulu |
| hye - Armenian | plt - Plateau Malagasy | |
| ibo - Igbo | pol - Polish | |

| | Set 1 | | | | | Set 2 | | | | | Set 3 | | | |
|---|---|---|---|---|---|---|---|---|---|---|---|---|---|---|
| **Lang** | **Acc** | **AUR.** | **ECE** | **Brier** | **Lang** | **Acc** | **AUR.** | **ECE** | **Brier** | **Lang** | **Acc** | **AUR.** | **ECE** | **Brier** |
| acm | 58.2 | 78.4 | 7.1 | 19.2 | arz | 69.8 | 77.8 | 12.2 | 18.2 | ceb | 65.8 | 75.6 | 11.1 | 19.7 |
| fin | 80.8 | 75.6 | 6.0 | 14.1 | hin | 67.0 | 72.9 | 14.0 | 20.6 | ita | 86.0 | 79.6 | 7.4 | 10.8 |
| khm | 4.0 | 5.5 | 66.2 | 51.4 | lvs | 74.8 | 79.7 | 11.3 | 16.2 | npi | 59.8 | 74.8 | 5.1 | 20.0 |
| pol | 80.0 | 78.9 | 6.3 | 13.3 | slv | 81.0 | 76.6 | 8.7 | 13.8 | swe | 79.2 | 81.6 | 7.3 | 12.8 |
| tso | 34.0 | 62.5 | 3.4 | 21.5 | xho | 37.0 | 60.0 | 5.0 | 22.8 | afr | 80.8 | 81.2 | 8.3 | 12.7 |
| asm | 45.8 | 68.0 | 6.1 | 22.8 | ces | 82.2 | 80.2 | 7.8 | 11.8 | fra | 86.2 | 77.4 | 9.9 | 11.4 |
| hin | 57.0 | 72.1 | 4.7 | 21.1 | jav | 68.0 | 78.4 | 11.1 | 18.4 | kin | 34.8 | 63.3 | 4.7 | 21.3 |
| mal | 60.5 | 74.0 | 12.3 | 21.3 | npi | 32.5 | 59.9 | 6.3 | 21.4 | por | 86.2 | 79.8 | 5.6 | 10.2 |
| sna | 35.8 | 58.6 | 6.2 | 23.0 | swh | 67.0 | 75.0 | 8.1 | 19.1 | tur | 78.2 | 78.8 | 8.5 | 14.6 |
| yor | 31.2 | 61.1 | 4.4 | 20.4 | als | 73.5 | 78.0 | 7.2 | 16.2 | azj | 66.8 | 71.8 | 14.3 | 21.4 |
| ckb | 46.0 | 71.8 | 8.3 | 22.2 | fuv | 28.0 | 51.9 | 5.2 | 20.4 | hrv | 79.8 | 78.1 | 8.2 | 14.3 |
| jpn | 66.5 | 73.3 | 29.9 | 28.0 | kir | 63.0 | 73.8 | 21.9 | 24.7 | mar | 67.5 | 73.5 | 9.2 | 19.4 |
| nso | 37.8 | 60.3 | 3.2 | 22.7 | snd | 17.5 | 50.5 | 30.0 | 25.3 | tam | 65.5 | 73.4 | 14.5 | 21.5 |
| ukr | 84.2 | 77.4 | 12.1 | 12.8 | zho | 76.5 | 71.2 | 30.6 | 25.2 | amh | 34.8 | 63.2 | 5.0 | 21.5 |
| bam | 31.2 | 60.6 | 4.8 | 20.8 | dan | 79.8 | 78.8 | 8.2 | 13.9 | gaz | 31.8 | 53.1 | 2.4 | 21.7 |
| hun | 82.5 | 84.0 | 9.7 | 11.8 | kac | 30.2 | 61.8 | 3.5 | 20.5 | kor | 77.8 | 77.9 | 14.5 | 16.3 |
| mkd | 77.8 | 79.1 | 7.8 | 14.4 | nya | 32.0 | 60.7 | 3.3 | 21.1 | ron | 80.0 | 80.4 | 6.3 | 13.0 |
| som | 35.2 | 59.0 | 3.9 | 22.2 | tel | 59.5 | 73.7 | 7.2 | 20.4 | urd | 59.5 | 67.3 | 19.7 | 25.8 |
| zho | 81.2 | 68.9 | 24.4 | 21.1 | apc | 65.0 | 78.3 | 9.7 | 18.4 | ben | 65.5 | 72.6 | 10.1 | 20.3 |
| deu | 86.8 | 72.3 | 13.4 | 12.9 | grn | 39.8 | 65.5 | 6.6 | 22.4 | hye | 0.2 | 1.5 | 63.8 | 42.1 |
| kan | 58.5 | 72.1 | 5.2 | 20.9 | lao | 32.5 | 59.5 | 2.1 | 21.5 | mlt | 69.8 | 76.4 | 9.9 | 18.4 |
| ory | 55.8 | 70.0 | 16.4 | 24.7 | rus | 81.2 | 81.0 | 10.4 | 13.0 | sot | 32.8 | 57.5 | 2.6 | 21.8 |
| tgk | 63.8 | 70.1 | 12.5 | 22.1 | urd | 41.2 | 64.8 | 4.3 | 22.5 | zsm | 82.5 | 82.9 | 9.5 | 11.8 |
| arb | 79.5 | 74.2 | 11.8 | 15.5 | ben | 35.2 | 59.9 | 4.9 | 22.4 | ell | 80.5 | 80.6 | 10.1 | 13.7 |
| guj | 58.0 | 68.8 | 8.2 | 22.0 | ibo | 40.2 | 62.0 | 3.7 | 22.6 | kat | 1.5 | 6.0 | 68.4 | 51.6 |
| lin | 34.2 | 59.3 | 3.5 | 21.9 | mri | 35.5 | 63.3 | 4.3 | 21.5 | pan | 58.0 | 74.4 | 11.4 | 21.6 |
| shn | 16.8 | 47.9 | 15.1 | 16.9 | spa | 84.0 | 79.3 | 5.8 | 11.7 | tgl | 75.2 | 80.1 | 7.8 | 15.3 |
| uzn | 69.0 | 77.0 | 13.4 | 19.2 | zul | 36.5 | 59.3 | 5.1 | 22.8 | arb | 29.8 | 56.4 | 3.6 | 20.8 |
| bod | 29.0 | 60.2 | 6.8 | 20.3 | sun | 65.5 | 74.3 | 13.5 | 20.9 | hat | 55.8 | 72.9 | 4.6 | 20.9 |
| ilo | 54.0 | 69.8 | 9.2 | 22.6 | kaz | 63.5 | 75.6 | 19.3 | 23.0 | lit | 73.8 | 82.3 | 11.0 | 15.7 |
| mya | 0.8 | 7.2 | 72.5 | 55.7 | pbt | 47.5 | 67.7 | 3.1 | 22.6 | sin | 32.2 | 59.2 | 3.8 | 21.5 |
| srp | 83.2 | 77.5 | 13.0 | 13.7 | tha | 71.8 | 75.5 | 19.8 | 21.2 | vie | 83.5 | 78.4 | 8.7 | 12.4 |
| ars | 62.0 | 78.5 | 9.6 | 18.8 | bul | 80.8 | 77.7 | 13.6 | 14.6 | est | 71.8 | 78.2 | 4.2 | 16.4 |
| hau | 45.2 | 67.1 | 11.7 | 23.9 | ind | 81.8 | 75.6 | 6.4 | 13.4 | kea | 48.8 | 73.0 | 8.0 | 21.1 |
| lug | 35.5 | 57.6 | 2.5 | 22.5 | nld | 83.0 | 78.2 | 5.9 | 12.0 | pes | 79.2 | 77.8 | 8.5 | 14.4 |
| sin | 58.8 | 72.7 | 10.3 | 21.6 | ssw | 31.8 | 61.5 | 3.0 | 20.8 | tir | 28.0 | 57.7 | 4.9 | 19.7 |
| war | 62.2 | 74.7 | 12.1 | 21.0 | ary | 58.5 | 72.0 | 7.7 | 21.2 | cat | 86.2 | 84.8 | 9.4 | 10.2 |
| eus | 69.5 | 75.7 | 7.0 | 18.1 | heb | 77.2 | 76.9 | 17.2 | 17.8 | isl | 67.0 | 78.3 | 5.2 | 17.4 |
| khk | 48.0 | 70.2 | 10.0 | 22.9 | luo | 31.8 | 55.7 | 5.7 | 21.5 | nob | 79.2 | 77.8 | 9.5 | 14.7 |
| plt | 44.2 | 65.6 | 5.8 | 23.1 | slk | 82.0 | 76.9 | 4.7 | 13.0 | **eng** | **87.8** | **87.9** | **4.0** | **8.1** |
| tsn | 31.8 | 62.7 | 4.6 | 20.8 | wol | 33.0 | 53.2 | 5.7 | 22.3 | **Avg.** | **57.6** | **68.9** | **10.9** | **19.8** |

Table 8: Per-language performance on the belebele test set for the **LLaMA3** model, reporting AUROC, ECE, and Brier score. Each row is color-coded by language category, based on resource availability (high, medium, low) and script type (Latin vs. non-Latin). The categories are shaded with soft pastel colors: high-resource Latin (light blue), high-resource non-Latin (light pink), medium-resource Latin (light green), medium-resource non-Latin (lavender), low-resource Latin (cream), and low-resource non-Latin (tan). English line and the Average line is **bolded**.

| | Set 1 | | | | | Set 2 | | | | | Set 3 | | | |
|---|---|---|---|---|---|---|---|---|---|---|---|---|---|---|
| **Lang** | **Acc** | **AUR.** | **ECE** | **Brier** | **Lang** | **Acc** | **AUR.** | **ECE** | **Brier** | **Lang** | **Acc** | **AUR.** | **ECE** | **Brier** |
| acm | 64.7 | 77.4 | 25.6 | 26.8 | arz | 75.7 | 82.3 | 16.7 | 17.5 | ceb | 47.3 | 72.9 | 29.9 | 30.7 |
| fin | 63.0 | 64.7 | 17.1 | 25.2 | hin | 67.3 | 75.0 | 22.7 | 24.1 | ita | 80.3 | 80.1 | 12.7 | 14.4 |
| lvs | 55.3 | 68.6 | 24.1 | 28.4 | npi | 43.3 | 67.7 | 32.7 | 33.4 | pol | 79.7 | 80.0 | 14.5 | 16.1 |
| slv | 63.3 | 75.4 | 23.5 | 25.9 | swe | 74.3 | 69.2 | 18.5 | 20.8 | tso | 34.3 | 57.7 | 29.4 | 32.3 |
| xho | 34.0 | 59.0 | 25.2 | 29.4 | afr | 71.0 | 80.5 | 18.7 | 20.1 | asm | 38.3 | 59.8 | 25.7 | 30.2 |
| ces | 80.7 | 81.5 | 13.4 | 14.2 | fra | 86.7 | 82.0 | 8.5 | 10.3 | hin | 54.0 | 68.5 | 28.6 | 30.4 |
| jav | 56.7 | 75.5 | 25.2 | 26.6 | kin | 36.7 | 58.0 | 26.8 | 30.7 | npi | 37.3 | 61.7 | 28.9 | 31.7 |
| por | 83.0 | 79.0 | 12.2 | 12.9 | sna | 34.0 | 66.5 | 28.5 | 29.1 | swh | 37.0 | 66.9 | 32.2 | 32.1 |
| tur | 78.3 | 82.0 | 12.9 | 15.6 | yor | 29.7 | 53.6 | 29.6 | 31.8 | als | 49.0 | 69.7 | 26.4 | 29.6 |
| azj | 58.7 | 67.0 | 18.4 | 26.0 | ckb | 47.0 | 63.6 | 29.4 | 33.0 | fuv | 27.7 | 54.9 | 34.1 | 34.0 |
| hrv | 68.0 | 76.6 | 21.4 | 23.1 | jpn | 78.3 | 75.8 | 13.1 | 16.1 | kir | 44.7 | 63.3 | 32.8 | 34.6 |
| mar | 45.3 | 66.6 | 32.3 | 33.9 | nso | 34.7 | 58.0 | 27.8 | 31.4 | snd | 40.7 | 62.8 | 22.8 | 28.6 |
| tam | 28.3 | 54.9 | 32.6 | 33.4 | ukr | 81.3 | 83.5 | 13.0 | 14.0 | zho | 84.3 | 86.8 | 8.6 | 10.6 |
| amh | 26.0 | 48.2 | 16.9 | 23.8 | bam | 35.3 | 62.1 | 25.8 | 29.2 | dan | 72.0 | 77.2 | 14.1 | 18.4 |
| gaz | 32.3 | 51.2 | 26.1 | 30.8 | hun | 60.0 | 72.2 | 23.7 | 26.2 | kac | 33.0 | 63.2 | 23.9 | 27.2 |
| kor | 78.7 | 77.2 | 14.7 | 15.8 | mkd | 61.0 | 74.9 | 22.7 | 25.1 | nya | 33.7 | 54.6 | 25.7 | 31.0 |
| ron | 82.0 | 82.7 | 12.8 | 13.5 | som | 33.0 | 63.9 | 29.7 | 30.6 | urd | 50.3 | 68.9 | 16.1 | 25.1 |
| zho | 80.3 | 76.2 | 11.6 | 14.9 | apc | 67.7 | 78.7 | 20.9 | 21.6 | ben | 49.7 | 65.0 | 20.7 | 28.1 |
| deu | 83.3 | 77.7 | 12.2 | 13.6 | grn | 35.7 | 64.7 | 31.9 | 32.3 | hye | 36.7 | 57.6 | 21.9 | 28.5 |
| mlt | 41.3 | 65.6 | 29.1 | 31.7 | rus | 83.3 | 82.0 | 12.4 | 13.3 | sot | 33.7 | 55.2 | 27.1 | 31.6 |
| tgk | 38.7 | 68.5 | 23.7 | 27.3 | urd | 41.3 | 63.0 | 26.0 | 30.9 | zsm | 79.7 | 81.3 | 12.3 | 14.4 |
| arb | 80.0 | 78.9 | 14.8 | 15.3 | ben | 36.0 | 62.8 | 29.3 | 31.5 | ell | 81.7 | 84.9 | 10.1 | 12.7 |
| wol | 31.6 | 56.5 | 22.0 | 27.7 | ibo | 30.0 | 50.7 | 27.4 | 31.9 | kat | 46.7 | 64.8 | 26.8 | 30.1 |
| lin | 33.0 | 63.0 | 34.9 | 34.2 | mri | 35.3 | 60.0 | 26.7 | 30.2 | spa | 79.0 | 78.2 | 19.1 | 18.3 |
| tgl | 69.0 | 73.6 | 17.5 | 21.6 | uzn | 46.9 | 66.0 | 29.1 | 32.3 | zul | 31.1 | 58.1 | 25.6 | 28.7 |
| arb | 28.3 | 54.7 | 33.8 | 34.0 | plt | 35.2 | 58.9 | 21.5 | 27.2 | hat | 46.4 | 67.9 | 27.2 | 30.3 |
| ilo | 37.8 | 68.2 | 30.7 | 31.1 | kaz | 42.1 | 67.3 | 30.4 | 32.3 | lit | 63.5 | 74.2 | 23.5 | 25.8 |
| pbt | 36.0 | 60.6 | 31.6 | 33.3 | sin | 36.0 | 58.2 | 23.0 | 29.0 | srp | 66.1 | 71.9 | 17.8 | 23.3 |
| tha | 49.0 | 69.2 | 24.9 | 28.7 | vie | 82.7 | 80.8 | 10.7 | 12.8 | ars | 65.6 | 78.3 | 25.7 | 25.6 |
| bul | 67.6 | 77.4 | 21.8 | 23.4 | est | 51.8 | 67.8 | 25.1 | 29.4 | hau | 29.3 | 60.8 | 33.2 | 32.3 |
| ind | 80.6 | 81.0 | 13.8 | 14.9 | kea | 47.5 | 67.2 | 33.1 | 34.4 | lug | 31.1 | 55.8 | 23.6 | 27.9 |
| nld | 81.1 | 79.4 | 12.6 | 14.7 | pes | 80.1 | 78.7 | 12.4 | 15.0 | tsn | 33.7 | 60.9 | 23.3 | 28.0 |
| ssw | 36.2 | 55.7 | 21.1 | 29.0 | tir | 27.6 | 54.7 | 23.9 | 27.0 | war | 49.2 | 66.5 | 26.3 | 30.6 |
| ary | 64.3 | 77.2 | 24.2 | 25.4 | cat | 76.5 | 81.4 | 14.7 | 16.3 | eus | 47.2 | 64.8 | 22.6 | 29.1 |
| heb | 79.8 | 79.6 | 11.5 | 14.7 | isl | 51.5 | 70.9 | 27.0 | 29.6 | khk | 37.5 | 57.6 | 21.7 | 28.8 |
| luo | 29.1 | 63.9 | 26.3 | 26.5 | nob | 72.7 | 77.0 | 16.2 | 19.0 | **eng** | **87.0** | **83.6** | **4.8** | **8.7** |
| slk | 76.8 | 80.7 | 15.1 | 16.9 | sun | 47.5 | 74.6 | 30.9 | 30.9 | **Avg.** | **53.0** | **68.0** | **22.8** | **25.2** |

Table 9: Per-language performance on the belebele test set for the **Aya** model, reporting AUROC, ECE, and Brier score. Each row is color-coded, same as Table 8. Language entries with lower than 5% accuracy is excluded. English line and the Average line is **bolded**.

| | Set 1 | | | | | Set 2 | | | | | Set 3 | | | |
|------|------|------|------|-------|------|------|------|------|-------|------|------|------|------|-------|
| **Lang** | **Acc** | **AUR.** | **ECE** | **Brier** | **Lang** | **Acc** | **AUR.** | **ECE** | **Brier** | **Lang** | **Acc** | **AUR.** | **ECE** | **Brier** |
| acm | 63.5 | 68.1 | 7.1 | 21.7 | arz | 72.2 | 77.2 | 7.1 | 16.6 | ceb | 52.8 | 74.2 | 17.2 | 23.6 |
| fin | 72.8 | 79.5 | 15.3 | 18.1 | hin | 67.5 | 68.1 | 9.2 | 20.8 | ita | 87.5 | 79.5 | 6.8 | 9.8 |
| khm | 10.8 | 57.5 | 21.5 | 15.0 | lvs | 73.5 | 71.5 | 10.7 | 18.1 | npi | 45.2 | 58.8 | 19.9 | 29.4 |
| pol | 81.8 | 82.8 | 9.0 | 12.2 | slv | 78.2 | 77.2 | 7.8 | 14.1 | swe | 83.8 | 76.8 | 8.0 | 12.3 |
| tso | 17.2 | 79.5 | 24.3 | 17.9 | xho | 34.8 | 57.0 | 27.5 | 31.4 | afr | 86.5 | 75.0 | 7.1 | 11.1 |
| asm | 52.0 | 66.5 | 19.6 | 27.1 | ces | 86.2 | 79.7 | 5.7 | 10.0 | fra | 89.5 | 86.4 | 6.3 | 8.0 |
| hin | 63.2 | 59.0 | 15.1 | 25.4 | jav | 53.8 | 68.6 | 10.8 | 23.5 | kin | 34.8 | 55.0 | 17.7 | 27.0 |
| mal | 54.0 | 62.0 | 15.3 | 26.4 | npi | 39.5 | 50.3 | 25.3 | 32.5 | por | 87.0 | 80.4 | 8.2 | 10.2 |
| sna | 21.5 | 62.7 | 22.8 | 22.1 | swh | 41.5 | 61.6 | 20.5 | 28.1 | tur | 78.2 | 74.0 | 9.8 | 15.2 |
| yor | 22.5 | 53.9 | 18.5 | 22.8 | als | 61.0 | 70.9 | 15.7 | 23.3 | azj | 63.8 | 65.8 | 15.1 | 23.8 |
| ckb | 5.0 | 47.7 | 38.9 | 21.6 | fuv | 29.2 | 60.4 | 21.5 | 26.1 | hrv | 80.5 | 77.8 | 8.1 | 13.6 |
| jpn | 76.2 | 87.5 | 7.5 | 11.7 | kir | 43.8 | 72.7 | 5.4 | 21.1 | mar | 59.5 | 65.0 | 12.7 | 24.4 |
| nso | 24.2 | 71.6 | 19.2 | 20.3 | snd | 26.0 | 63.9 | 10.6 | 19.3 | tam | 50.8 | 69.7 | 13.6 | 23.9 |
| ukr | 82.0 | 73.0 | 5.6 | 13.0 | zho | 86.5 | 86.6 | 2.6 | 8.5 | amh | 22.5 | 68.0 | 6.4 | 17.3 |
| bam | 28.8 | 67.5 | 20.7 | 23.6 | dan | 85.8 | 77.5 | 6.8 | 11.0 | gaz | 29.5 | 54.5 | 23.5 | 27.4 |
| hun | 72.2 | 74.9 | 15.7 | 19.6 | kac | 29.8 | 52.3 | 14.5 | 24.4 | kor | 82.8 | 76.9 | 4.3 | 12.0 |
| mkd | 76.8 | 75.3 | 6.2 | 15.7 | nya | 25.0 | 64.9 | 26.1 | 25.3 | ron | 81.8 | 77.5 | 7.8 | 13.1 |
| som | 29.8 | 59.0 | 25.1 | 27.7 | tel | 42.2 | 66.2 | 29.5 | 31.1 | urd | 63.8 | 67.8 | 6.5 | 21.1 |
| zho | 86.8 | 79.6 | 4.6 | 8.8 | apc | 70.2 | 69.1 | 6.9 | 19.4 | ben | 65.5 | 70.7 | 19.6 | 23.9 |
| deu | 90.2 | 80.2 | 4.8 | 7.6 | grn | 35.2 | 63.1 | 21.8 | 26.9 | hye | 23.0 | 63.1 | 8.8 | 19.0 |
| kan | 46.2 | 61.9 | 25.9 | 31.4 | lao | 5.5 | 62.6 | 28.6 | 15.3 | mlt | 44.2 | 67.2 | 25.3 | 29.1 |
| ory | 50.8 | 66.1 | 21.5 | 27.9 | rus | 86.8 | 79.0 | 5.6 | 10.0 | sot | 31.2 | 53.7 | 20.7 | 28.0 |
| tgk | 40.0 | 64.2 | 16.6 | 26.0 | urd | 47.5 | 64.1 | 21.9 | 28.7 | zsm | 80.2 | 76.4 | 4.2 | 12.8 |
| arb | 85.2 | 76.1 | 4.8 | 10.8 | ben | 33.0 | 57.7 | 27.9 | 30.6 | ell | 74.5 | 74.4 | 11.8 | 17.3 |
| guj | 55.2 | 67.4 | 12.7 | 24.1 | ibo | 18.8 | 54.1 | 29.5 | 26.1 | kat | 15.0 | 63.1 | 22.9 | 19.6 |
| lin | 28.8 | 57.3 | 32.7 | 32.5 | mri | 12.5 | 87.2 | 25.1 | 14.8 | pan | 50.0 | 65.7 | 18.8 | 26.8 |
| shn | 1.8 | 39.6 | 27.4 | 10.0 | spa | 88.5 | 85.2 | 6.2 | 8.4 | tgl | 68.5 | 74.0 | 9.5 | 19.4 |
| uzn | 59.8 | 70.2 | 11.8 | 22.3 | zul | 27.0 | 55.4 | 31.6 | 31.1 | arb | 31.5 | 68.0 | 16.8 | 22.6 |
| bod | 25.0 | 57.0 | 20.0 | 25.2 | sun | 49.8 | 64.2 | 17.0 | 26.9 | hat | 43.5 | 65.5 | 29.1 | 31.3 |
| ilo | 34.5 | 62.4 | 24.1 | 28.0 | kaz | 48.8 | 67.3 | 13.1 | 25.0 | lit | 68.5 | 71.3 | 14.3 | 20.8 |
| mya | 8.0 | 68.6 | 19.2 | 10.6 | pbt | 18.8 | 50.1 | 26.5 | 24.1 | sin | 30.5 | 61.6 | 19.2 | 24.6 |
| srp | 82.8 | 76.0 | 5.5 | 12.3 | tha | 48.0 | 72.9 | 3.8 | 21.0 | vie | 84.2 | 84.8 | 4.8 | 9.8 |
| ars | 71.8 | 72.6 | 3.1 | 17.6 | bul | 78.5 | 81.3 | 4.2 | 12.8 | est | 65.5 | 68.7 | 16.2 | 22.9 |
| hau | 26.5 | 54.9 | 30.1 | 30.4 | ind | 80.2 | 80.4 | 7.3 | 13.1 | kea | 47.2 | 63.1 | 30.1 | 33.2 |
| lug | 27.8 | 58.1 | 24.6 | 26.8 | nld | 84.5 | 83.2 | 9.6 | 11.2 | pes | 69.0 | 72.2 | 4.1 | 18.4 |
| sin | 25.2 | 69.1 | 27.2 | 24.2 | ssw | 25.2 | 65.6 | 23.8 | 23.8 | tir | 18.8 | 64.7 | 5.3 | 15.3 |
| war | 53.2 | 71.0 | 14.5 | 23.9 | ary | 63.0 | 66.1 | 8.3 | 22.1 | cat | 86.0 | 80.4 | 7.0 | 10.6 |
| eus | 46.2 | 64.8 | 18.0 | 26.8 | heb | 78.0 | 76.8 | 6.4 | 14.6 | isl | 61.0 | 68.3 | 17.6 | 24.9 |
| khk | 33.8 | 66.4 | 14.9 | 23.8 | luo | 30.8 | 57.1 | 21.9 | 27.1 | nob | 80.2 | 79.8 | 7.2 | 12.9 |
| plt | 35.5 | 64.4 | 8.5 | 22.5 | slk | 81.8 | 82.3 | 9.2 | 12.7 | **eng** | **91.8** | **83.9** | **4.2** | **6.7** |
| tsn | 28.8 | 62.0 | 23.6 | 26.3 | wol | 31.0 | 51.9 | 23.1 | 29.0 | **Avg.** | **52.7** | **68.6** | **15.2** | **20.4** |

Table 10: Per-language performance on the belebele test set for the **Qwen 2.5** model, reporting AUROC, ECE, and Brier score. Each row is color-coded, same as Table 8. English line and the Average line is **bolded**.

| | Set 1 | | | | | Set 2 | | | | | Set 3 | | | |
|---|---|---|---|---|---|---|---|---|---|---|---|---|---|---|
| **Lang** | **Acc** | **AUR.** | **ECE** | **Brier** | **Lang** | **Acc** | **AUR.** | **ECE** | **Brier** | **Lang** | **Acc** | **AUR.** | **ECE** | **Brier** |
| acm | 36.0 | 60.5 | 17.8 | 22.8 | arz | 38.5 | 65.0 | 11.6 | 22.6 | ceb | 36.0 | 60.4 | 19.9 | 22.9 |
| fin | 41.5 | 70.9 | 8.3 | 21.5 | hin | 41.8 | 60.8 | 11.8 | 23.7 | ita | 70.5 | 73.2 | 15.3 | 20.3 |
| khm | 20.2 | 53.7 | 29.7 | 18.9 | lvs | 40.2 | 68.6 | 16.4 | 22.2 | npi | 34.5 | 59.6 | 13.7 | 22.8 |
| pol | 58.8 | 71.5 | 9.8 | 21.8 | slv | 46.8 | 67.3 | 4.5 | 22.8 | swe | 58.8 | 76.7 | 12.2 | 20.4 |
| tso | 27.5 | 52.8 | 29.1 | 20.9 | xho | 26.2 | 51.8 | 31.1 | 21.1 | afr | 53.2 | 65.8 | 8.1 | 23.4 |
| asm | 26.8 | 59.6 | 20.7 | 19.6 | ces | 66.2 | 67.1 | 10.7 | 21.7 | fra | 69.5 | 75.0 | 15.1 | 19.8 |
| hin | 31.2 | 60.1 | 17.8 | 22.7 | jav | 39.2 | 61.0 | 13.4 | 22.9 | kin | 27.5 | 54.3 | 26.5 | 20.9 |
| mal | 28.2 | 60.4 | 23.5 | 21.0 | npi | 28.2 | 47.9 | 37.3 | 23.3 | por | 65.5 | 75.2 | 13.3 | 20.2 |
| sna | 24.0 | 50.2 | 35.5 | 20.2 | swh | 31.8 | 61.5 | 15.7 | 22.7 | tur | 47.2 | 63.1 | 11.7 | 23.9 |
| yor | 25.0 | 51.7 | 29.7 | 19.8 | als | 43.8 | 62.6 | 10.3 | 23.6 | azj | 33.0 | 60.5 | 20.1 | 22.0 |
| ckb | 28.8 | 51.9 | 27.4 | 21.6 | fuv | 10.5 | 48.4 | 39.0 | 13.7 | hrv | 51.0 | 67.9 | 4.1 | 22.4 |
| jpn | 54.2 | 66.3 | 12.0 | 23.8 | kir | 24.2 | 54.6 | 36.7 | 21.2 | mar | 37.0 | 57.1 | 11.9 | 23.5 |
| nso | 26.5 | 56.0 | 24.8 | 20.2 | snd | 13.5 | 44.3 | 38.7 | 18.0 | tam | 32.5 | 59.1 | 30.7 | 23.5 |
| ukr | 62.0 | 71.8 | 19.2 | 21.9 | zho | 25.8 | 53.0 | 26.1 | 23.9 | amh | 25.5 | 53.4 | 34.9 | 19.8 |
| bam | 28.2 | 54.7 | 28.1 | 20.6 | dan | 53.0 | 74.8 | 7.9 | 20.8 | gaz | 25.8 | 49.2 | 28.6 | 20.4 |
| hun | 54.8 | 70.6 | 6.7 | 21.9 | kac | 25.2 | 49.0 | 36.7 | 20.5 | kor | 55.0 | 65.4 | 11.0 | 24.0 |
| mkd | 50.0 | 68.4 | 9.8 | 22.6 | nya | 27.8 | 47.5 | 33.8 | 22.0 | ron | 35.2 | 57.9 | 25.6 | 26.0 |
| som | 25.2 | 53.4 | 27.3 | 20.1 | tel | 27.8 | 53.8 | 29.4 | 22.2 | urd | 17.2 | 47.7 | 42.8 | 23.4 |
| zho | 54.2 | 73.3 | 23.8 | 26.5 | apc | 37.0 | 64.8 | 11.6 | 22.1 | ben | 31.2 | 58.2 | 23.5 | 21.7 |
| deu | 65.8 | 74.5 | 11.9 | 20.0 | grn | 31.0 | 58.7 | 18.9 | 21.8 | hye | 17.0 | 48.7 | 31.0 | 16.8 |
| kan | 27.8 | 57.9 | 23.1 | 23.5 | lao | 27.0 | 44.3 | 40.1 | 21.3 | mlt | 33.0 | 61.4 | 15.0 | 21.4 |
| ory | 30.2 | 52.5 | 22.0 | 21.5 | rus | 59.2 | 72.9 | 19.9 | 23.4 | sot | 23.2 | 52.5 | 34.9 | 19.9 |
| tgk | 26.5 | 52.2 | 38.7 | 20.9 | urd | 26.5 | 56.6 | 34.0 | 20.6 | zsm | 56.5 | 71.0 | 17.2 | 23.9 |
| arb | 47.5 | 67.2 | 15.2 | 23.3 | ben | 28.0 | 50.0 | 36.6 | 21.8 | ell | 54.5 | 75.0 | 9.0 | 20.6 |
| guj | 28.5 | 53.0 | 31.6 | 22.7 | ibo | 26.0 | 52.3 | 31.8 | 21.0 | kat | 27.5 | 59.1 | 23.8 | 21.6 |
| lin | 27.5 | 54.0 | 29.1 | 20.9 | mri | 28.8 | 54.1 | 35.0 | 21.6 | pan | 29.5 | 54.1 | 31.3 | 22.1 |
| spa | 68.2 | 67.4 | 22.6 | 23.3 | tgl | 41.2 | 64.8 | 14.9 | 22.7 | tsn | 26.8 | 55.7 | 26.8 | 20.6 |
| uzn | 35.0 | 59.7 | 24.4 | 22.7 | zul | 27.2 | 51.5 | 19.6 | 20.9 | plt | 31.8 | 58.2 | 20.5 | 21.7 |
| bod | 27.2 | 45.7 | 33.7 | 22.1 | hat | 30.8 | 52.8 | 19.5 | 22.5 | wol | 27.2 | 52.2 | 28.0 | 21.4 |
| ilo | 30.2 | 57.3 | 22.4 | 21.4 | kaz | 29.5 | 50.2 | 35.4 | 22.6 | lit | 42.8 | 65.7 | 18.5 | 23.1 |
| pbt | 22.8 | 57.5 | 24.8 | 18.7 | sin | 24.5 | 51.0 | 46.5 | 20.3 | srp | 54.5 | 70.4 | 16.7 | 23.2 |
| tha | 48.5 | 66.7 | 11.6 | 23.5 | vie | 59.5 | 73.6 | 11.4 | 21.2 | ars | 38.0 | 62.4 | 17.3 | 23.1 |
| bul | 49.5 | 71.8 | 14.5 | 22.3 | est | 37.0 | 60.7 | 14.0 | 23.4 | hau | 25.0 | 53.5 | 34.0 | 19.9 |
| ind | 60.8 | 73.8 | 17.7 | 21.7 | kea | 37.2 | 59.3 | 15.2 | 23.6 | lug | 28.2 | 48.7 | 24.4 | 21.6 |
| nld | 62.5 | 75.5 | 15.7 | 20.7 | pes | 57.2 | 65.7 | 14.7 | 23.7 | sin | 32.2 | 54.6 | 28.4 | 22.4 |
| ssw | 25.2 | 54.6 | 25.5 | 20.0 | tir | 23.8 | 49.8 | 34.9 | 19.0 | war | 36.0 | 58.3 | 20.6 | 22.8 |
| ary | 30.0 | 61.2 | 20.9 | 21.7 | cat | 65.0 | 71.1 | 11.9 | 21.6 | eus | 37.8 | 64.6 | 10.7 | 22.3 |
| heb | 39.5 | 68.8 | 16.9 | 21.7 | isl | 34.8 | 56.0 | 17.3 | 23.6 | khk | 27.0 | 56.6 | 26.3 | 20.4 |
| luo | 26.2 | 57.1 | 23.8 | 19.6 | nob | 53.2 | 70.0 | 13.3 | 22.3 | **eng** | **73.5** | **79.5** | **10.5** | **16.3** |
| slk | 56.8 | 72.3 | 16.6 | 21.5 | sun | 36.0 | 57.3 | 12.7 | 22.8 | **Avg.** | **37.3** | **59.7** | **22.2** | **21.7** |

Table 11: Per-language performance on the belebele test set for the **Deepseek** model, reporting AUROC, ECE, and Brier score. Each row is color-coded, same as Table 8. Language entries with lower than 5% accuracy is excluded. English line and the Average line is **bolded**.

| | Set 1 | | | | | Set 2 | | | | | Set 3 | | | |
|---|---|---|---|---|---|---|---|---|---|---|---|---|---|---|
| **Lang** | **Acc** | **AUR.** | **ECE** | **Brier** | **Lang** | **Acc** | **AUR.** | **ECE** | **Brier** | **Lang** | **Acc** | **AUR.** | **ECE** | **Brier** |
| acm | 21.2 | 77.2 | 10.3 | 15.3 | arz | 17.5 | 74.5 | 14.0 | 14.1 | ceb | 39.5 | 64.9 | 22.2 | 27.9 |
| fin | 53.0 | 69.5 | 17.4 | 25.3 | hin | 5.0 | 68.6 | 18.5 | 8.6 | ita | 73.2 | 75.3 | 11.4 | 17.5 |
| pol | 61.2 | 71.0 | 9.9 | 21.4 | slv | 66.0 | 68.4 | 9.7 | 21.1 | swe | 71.8 | 72.7 | 7.1 | 17.8 |
| tso | 26.5 | 54.5 | 20.2 | 24.7 | xho | 26.5 | 55.1 | 22.5 | 25.9 | afr | 60.2 | 69.5 | 14.3 | 23.3 |
| ces | 65.0 | 69.5 | 6.4 | 20.5 | fra | 74.8 | 71.6 | 5.1 | 16.6 | lvs | 38.5 | 61.9 | 13.2 | 25.6 |
| hin | 38.0 | 63.0 | 20.6 | 27.0 | jav | 35.2 | 58.2 | 21.1 | 28.0 | kin | 26.5 | 53.0 | 21.8 | 25.4 |
| npi | 29.0 | 47.2 | 18.8 | 26.6 | por | 75.8 | 73.5 | 7.8 | 16.3 | sun | 30.5 | 53.0 | 21.1 | 28.0 |
| sna | 29.5 | 54.0 | 20.4 | 26.4 | swh | 29.0 | 60.5 | 29.4 | 29.6 | tur | 27.0 | 62.1 | 22.8 | 25.8 |
| yor | 22.0 | 51.6 | 15.7 | 21.4 | als | 35.2 | 64.0 | 16.9 | 25.1 | azj | 19.0 | 54.4 | 22.6 | 24.2 |
| jpn | 21.2 | 71.5 | 13.9 | 17.2 | kir | 30.0 | 57.9 | 7.2 | 22.9 | mar | 5.5 | 56.6 | 17.3 | 8.9 |
| nso | 26.0 | 49.4 | 18.7 | 24.8 | tam | 8.8 | 48.5 | 21.5 | 14.6 | ukr | 69.8 | 66.8 | 5.6 | 19.8 |
| zho | 54.0 | 81.2 | 4.2 | 17.6 | ssw | 25.5 | 54.9 | 20.2 | 23.8 | fuv | 20.2 | 42.9 | 20.0 | 23.2 |
| bam | 24.0 | 57.5 | 21.5 | 25.3 | dan | 72.2 | 68.9 | 7.8 | 18.5 | gaz | 23.8 | 57.0 | 20.2 | 22.8 |
| hun | 61.2 | 71.1 | 11.3 | 21.4 | kac | 27.2 | 48.0 | 16.9 | 23.7 | kor | 25.8 | 60.3 | 9.5 | 20.1 |
| mkd | 54.2 | 67.3 | 7.3 | 23.2 | nya | 28.5 | 55.2 | 20.0 | 25.6 | ron | 68.2 | 72.5 | 6.7 | 18.5 |
| som | 26.0 | 54.7 | 20.6 | 25.0 | zho | 49.5 | 81.0 | 5.8 | 17.9 | apc | 18.5 | 69.3 | 13.1 | 15.4 |
| deu | 70.2 | 70.8 | 8.1 | 18.9 | grn | 30.8 | 53.0 | 19.9 | 27.6 | mlt | 29.2 | 58.1 | 17.1 | 24.8 |
| rus | 72.0 | 70.2 | 5.6 | 18.1 | sot | 25.5 | 52.4 | 17.1 | 23.0 | tgk | 22.0 | 48.7 | 9.9 | 20.4 |
| urd | 29.0 | 57.7 | 20.4 | 25.3 | zsm | 58.8 | 71.1 | 12.7 | 22.8 | arb | 27.5 | 78.2 | 8.0 | 16.2 |
| ben | 30.0 | 55.0 | 21.3 | 26.4 | hrv | 68.2 | 72.2 | 9.8 | 19.8 | ell | 6.8 | 46.5 | 32.2 | 20.2 |
| ibo | 18.8 | 42.1 | 28.2 | 27.9 | lin | 27.2 | 49.5 | 19.3 | 25.8 | mri | 26.8 | 45.9 | 21.4 | 26.2 |
| spa | 72.8 | 70.6 | 6.3 | 18.1 | tgl | 49.0 | 71.2 | 17.7 | 24.9 | uzn | 30.2 | 64.8 | 20.9 | 24.8 |
| zul | 22.2 | 51.6 | 23.5 | 24.4 | arb | 17.2 | 46.8 | 14.5 | 19.4 | hat | 33.5 | 58.4 | 23.9 | 28.7 |
| kaz | 26.5 | 51.4 | 10.1 | 22.0 | lit | 41.0 | 64.2 | 13.8 | 25.2 | sin | 26.2 | 51.1 | 13.9 | 23.1 |
| srp | 68.5 | 68.3 | 7.5 | 20.2 | tha | 11.8 | 54.6 | 14.3 | 14.0 | vie | 27.5 | 50.6 | 39.2 | 38.0 |
| ars | 19.2 | 82.4 | 12.3 | 13.4 | bul | 67.8 | 71.5 | 5.5 | 19.2 | est | 43.8 | 59.5 | 16.0 | 27.6 |
| hau | 21.5 | 46.5 | 22.0 | 25.9 | ind | 62.8 | 72.5 | 12.3 | 21.7 | kea | 35.8 | 59.0 | 23.6 | 29.5 |
| lug | 26.0 | 57.2 | 23.7 | 25.5 | nld | 68.5 | 74.0 | 13.0 | 19.5 | ilo | 28.5 | 56.4 | 24.4 | 27.5 |
| war | 35.5 | 61.5 | 27.5 | 30.7 | ary | 16.2 | 74.8 | 12.6 | 13.5 | cat | 71.2 | 75.8 | 10.1 | 17.6 |
| eus | 30.0 | 53.8 | 17.6 | 25.1 | heb | 10.0 | 62.1 | 26.7 | 17.0 | isl | 23.2 | 59.4 | 28.8 | 27.5 |
| khk | 23.5 | 51.3 | 8.9 | 21.2 | luo | 25.8 | 48.4 | 20.1 | 25.0 | **eng** | **80.8** | **83.3** | **10.6** | **13.3** |
| tsn | 27.2 | 47.3 | 20.1 | 26.4 | wol | 22.8 | 47.7 | 20.4 | 24.2 | **Avg.** | **32.2** | **56.6** | **19.8** | **22.1** |

Table 12: Per-language performance on the belebele test set for the **Mistral** model, reporting AUROC, ECE, and Brier score. Each row is color-coded, same as Table 8. Language entries with lower than 5% accuracy is excluded. English line and the Average line is **bolded**.

| | Set 1 | | | | | Set 2 | | | | | Set 3 | | | |
|------|------|------|------|-------|------|------|------|------|-------|------|------|------|------|-------|
| Lang | Acc | AUR. | ECE | Brier | Lang | Acc | AUR. | ECE | Brier | Lang | Acc | AUR. | ECE | Brier |
| acm | 73.0 | 81.3 | 6.1 | 15.2 | arz | 82.5 | 85.5 | 5.1 | 10.8 | ceb | 69.2 | 79.3 | 6.9 | 16.5 |
| fin | 89.5 | 85.4 | 4.1 | 7.4 | hin | 79.0 | 81.2 | 5.7 | 13.0 | ita | 89.5 | 92.2 | 6.0 | 6.4 |
| lvs | 84.8 | 87.8 | 8.6 | 9.5 | npi | 73.0 | 80.8 | 9.1 | 15.2 | pol | 90.5 | 88.9 | 7.2 | 6.6 |
| slv | 90.5 | 91.3 | 7.0 | 6.2 | swe | 89.8 | 86.8 | 6.6 | 6.9 | tso | 37.2 | 62.0 | 6.0 | 22.4 |
| xho | 38.0 | 65.2 | 8.5 | 22.1 | afr | 91.2 | 82.9 | 6.7 | 6.7 | ces | 90.0 | 90.0 | 9.3 | 7.0 |
| fra | 93.5 | 88.9 | 11.2 | 5.6 | hin | 67.2 | 73.7 | 3.8 | 18.8 | jav | 78.8 | 82.0 | 3.3 | 12.4 |
| kin | 33.2 | 63.4 | 6.3 | 21.3 | por | 93.2 | 93.0 | 7.9 | 5.0 | sna | 36.8 | 62.6 | 6.3 | 22.6 |
| swh | 79.0 | 78.6 | 4.2 | 13.2 | tur | 87.0 | 86.0 | 5.6 | 8.7 | yor | 37.0 | 56.6 | 8.9 | 23.8 |
| als | 83.2 | 88.0 | 5.3 | 9.8 | azj | 72.8 | 78.9 | 3.4 | 15.8 | fuv | 27.0 | 54.6 | 9.6 | 21.3 |
| hrv | 89.5 | 89.0 | 5.1 | 6.9 | jpn | 85.5 | 84.9 | 12.2 | 10.0 | kir | 69.8 | 78.0 | 7.3 | 17.4 |
| nso | 35.8 | 60.9 | 5.8 | 22.6 | tam | 75.0 | 78.2 | 4.2 | 14.9 | ukr | 91.5 | 90.0 | 5.8 | 6.2 |
| zho | 88.2 | 92.7 | 11.3 | 7.5 | bam | 35.5 | 58.6 | 5.1 | 22.5 | dan | 92.0 | 87.5 | 6.4 | 5.7 |
| gaz | 30.8 | 52.5 | 7.0 | 22.0 | hun | 90.0 | 85.5 | 3.7 | 6.5 | kac | 32.5 | 53.7 | 8.0 | 22.7 |
| kor | 88.5 | 83.2 | 7.7 | 8.2 | mkd | 86.0 | 88.6 | 6.9 | 8.9 | nya | 35.0 | 64.0 | 4.8 | 21.9 |
| ron | 90.0 | 89.1 | 5.5 | 6.7 | som | 28.2 | 54.0 | 15.7 | 24.0 | zho | 88.5 | 92.4 | 11.9 | 7.0 |
| apc | 77.2 | 83.4 | 7.3 | 13.6 | deu | 93.8 | 86.0 | 6.8 | 4.9 | grn | 35.2 | 66.0 | 14.7 | 22.8 |
| mlt | 63.5 | 73.2 | 7.0 | 19.9 | ory | 74.8 | 79.0 | 7.5 | 15.7 | rus | 91.8 | 92.0 | 4.7 | 5.3 |
| sot | 32.5 | 59.6 | 5.7 | 21.7 | tgk | 55.5 | 72.0 | 5.8 | 21.2 | urd | 46.2 | 65.8 | 2.6 | 23.0 |
| zsm | 88.8 | 85.1 | 4.4 | 7.5 | arb | 90.8 | 85.2 | 6.1 | 6.7 | ben | 34.0 | 59.8 | 12.8 | 23.0 |
| ell | 88.5 | 90.4 | 8.9 | 7.8 | wol | 32.7 | 52.8 | 9.5 | 24.0 | ibo | 28.2 | 57.5 | 11.0 | 21.4 |
| lin | 32.2 | 61.3 | 13.9 | 22.9 | mri | 37.0 | 60.0 | 8.3 | 23.3 | pan | 79.0 | 87.2 | 7.8 | 11.8 |
| spa | 91.3 | 90.5 | 5.8 | 6.1 | tgl | 83.3 | 83.4 | 5.1 | 11.1 | zul | 35.3 | 58.6 | 6.3 | 22.4 |
| hat | 58.7 | 78.3 | 9.5 | 19.1 | ilo | 43.3 | 73.9 | 7.7 | 20.7 | kaz | 64.7 | 82.3 | 9.4 | 16.7 |
| lit | 86.7 | 89.5 | 12.9 | 9.0 | arb | 26.0 | 61.4 | 7.9 | 19.0 | pbt | 59.3 | 66.3 | 12.5 | 22.4 |
| sin | 27.3 | 59.0 | 6.8 | 20.0 | srp | 90.7 | 95.1 | 8.0 | 5.5 | tha | 47.3 | 65.2 | 18.6 | 26.8 |
| vie | 92.0 | 93.1 | 8.6 | 5.4 | ars | 78.7 | 84.7 | 8.0 | 11.3 | bul | 89.3 | 89.0 | 9.8 | 6.9 |
| est | 79.3 | 82.4 | 4.2 | 12.8 | hau | 38.7 | 56.1 | 10.8 | 23.9 | ind | 88.7 | 93.6 | 7.0 | 6.9 |
| kea | 56.7 | 72.4 | 11.5 | 21.8 | lug | 24.7 | 55.1 | 16.1 | 22.1 | nld | 90.0 | 86.9 | 5.4 | 7.2 |
| ssw | 26.0 | 61.4 | 15.8 | 20.1 | war | 52.7 | 77.5 | 11.1 | 20.3 | ary | 64.0 | 80.4 | 10.4 | 17.3 |
| cat | 91.3 | 85.6 | 4.4 | 6.2 | eus | 70.0 | 75.9 | 6.8 | 17.3 | heb | 90.0 | 83.2 | 10.3 | 8.6 |
| isl | 83.3 | 78.6 | 3.3 | 11.1 | khk | 58.0 | 60.9 | 11.2 | 24.4 | luo | 30.7 | 52.1 | 12.8 | 23.5 |
| nob | 88.0 | 83.2 | 4.3 | 8.3 | plt | 57.3 | 81.3 | 11.2 | 17.8 | **eng** | **94.0** | **89.0** | **6.9** | **3.6** |
| sun | 50.7 | 72.8 | 11.6 | 21.2 | tsn | 30.7 | 57.9 | 10.7 | 22.4 | **Avg.** | **65.4** | **75.5** | **9.1** | **15.3** |

Table 13: Per-language performance on the belebele test set for the **Phi** model, reporting AUROC, ECE, and Brier score. Each row is color-coded, same as Table 8. Language entries with lower than 5% accuracy is excluded. English line and the Average line is **bolded**.

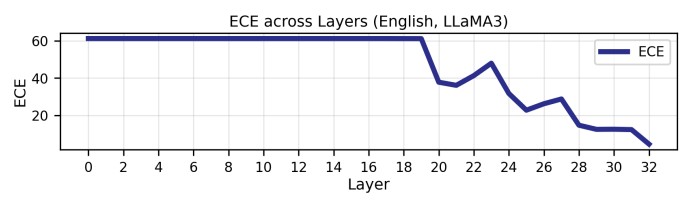

Figure 9: Layer-wise predicted confidence ECE for English in LLaMA3.

## C LAYER-WISE CALIBRATION ANALYSIS

### C.1 ENGLISH CALIBRATION IMPROVES AS LAYER DEEPENS

Figure 9 shows the layer-wise Expected Calibration Error (ECE) for English in LLaMA3, illustrating how calibration improves progressively in deeper layers.

### C.2 MULTILINGUAL CALIBRATION IS BEST AT LATE-INTERMEDIATE LAYERS

We visualize calibration performance across layers by plotting metrics against entropy on the MMMLU dataset in LLaMA3 (Figure 10), Cohere (Figure 11), Mistral (Figure 12), Phi (Figure 13), Deepseek (Figure 14), Qwen 2.5 (Figure 15).

### C.3 RELIABILITY DIAGRAMS

Figures 16 and 17 present reliability diagrams for LLaMA3 and Aya, respectively, illustrating calibration behaviour across languages and comparing intermediate versus final layers.

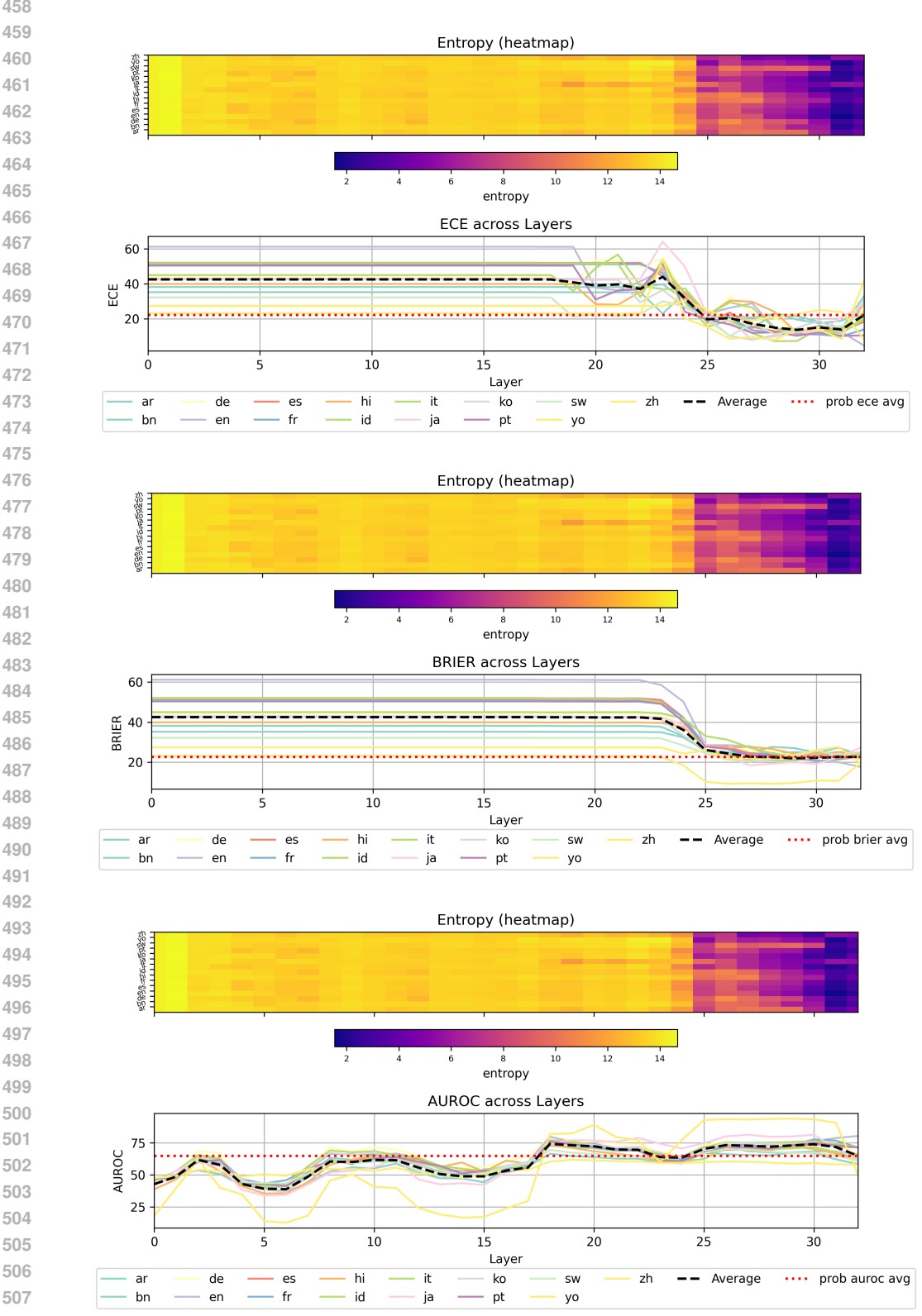

Figure 10: Calibration metrics (ECE, Brier score, AUROC) vs. entropy across layers on the MMMLU subset for LLaMA3.

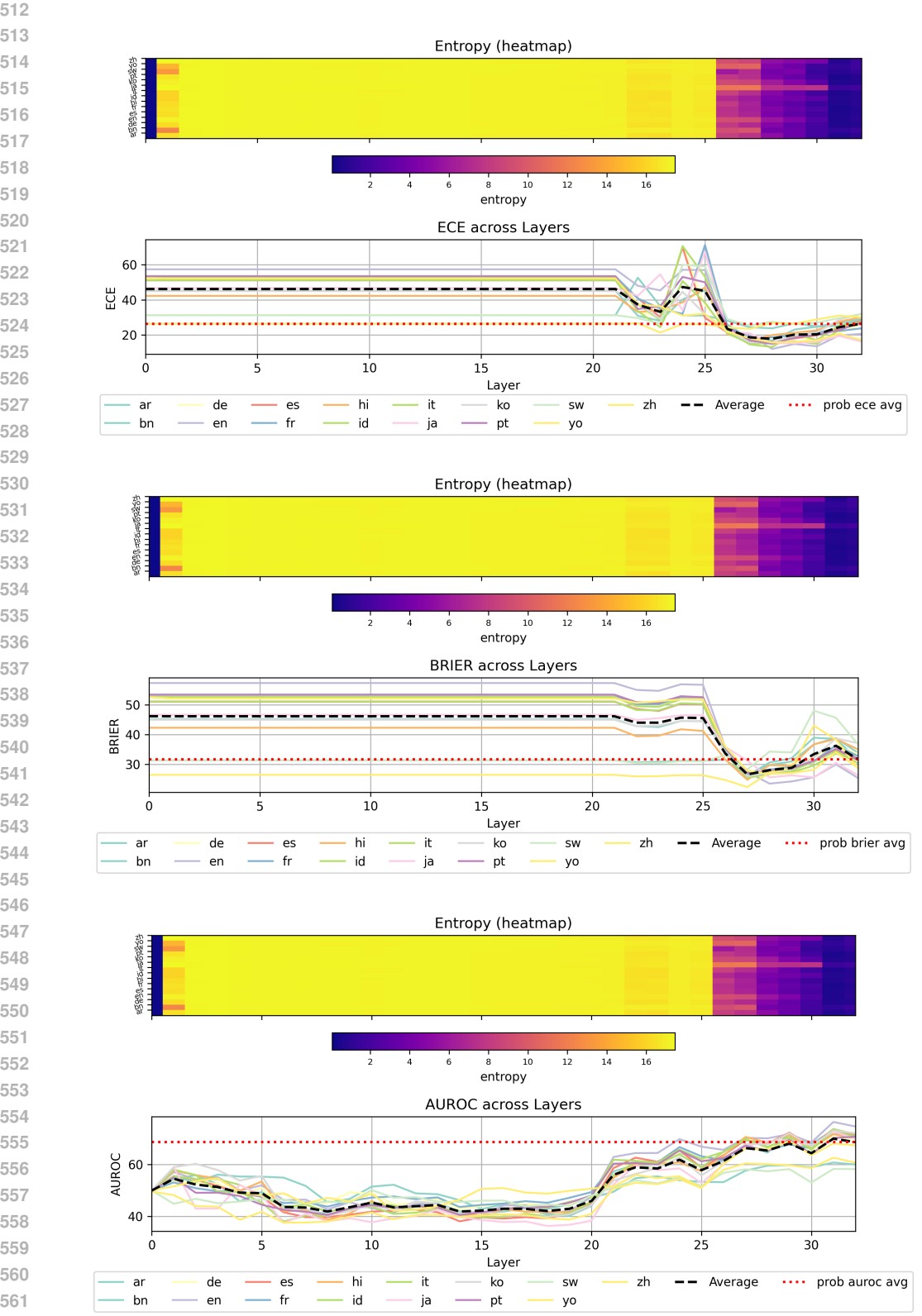

Figure 11: Calibration metrics (ECE, Brier score, AUROC) vs. entropy across layers on the MMMLU dataset for Aya.

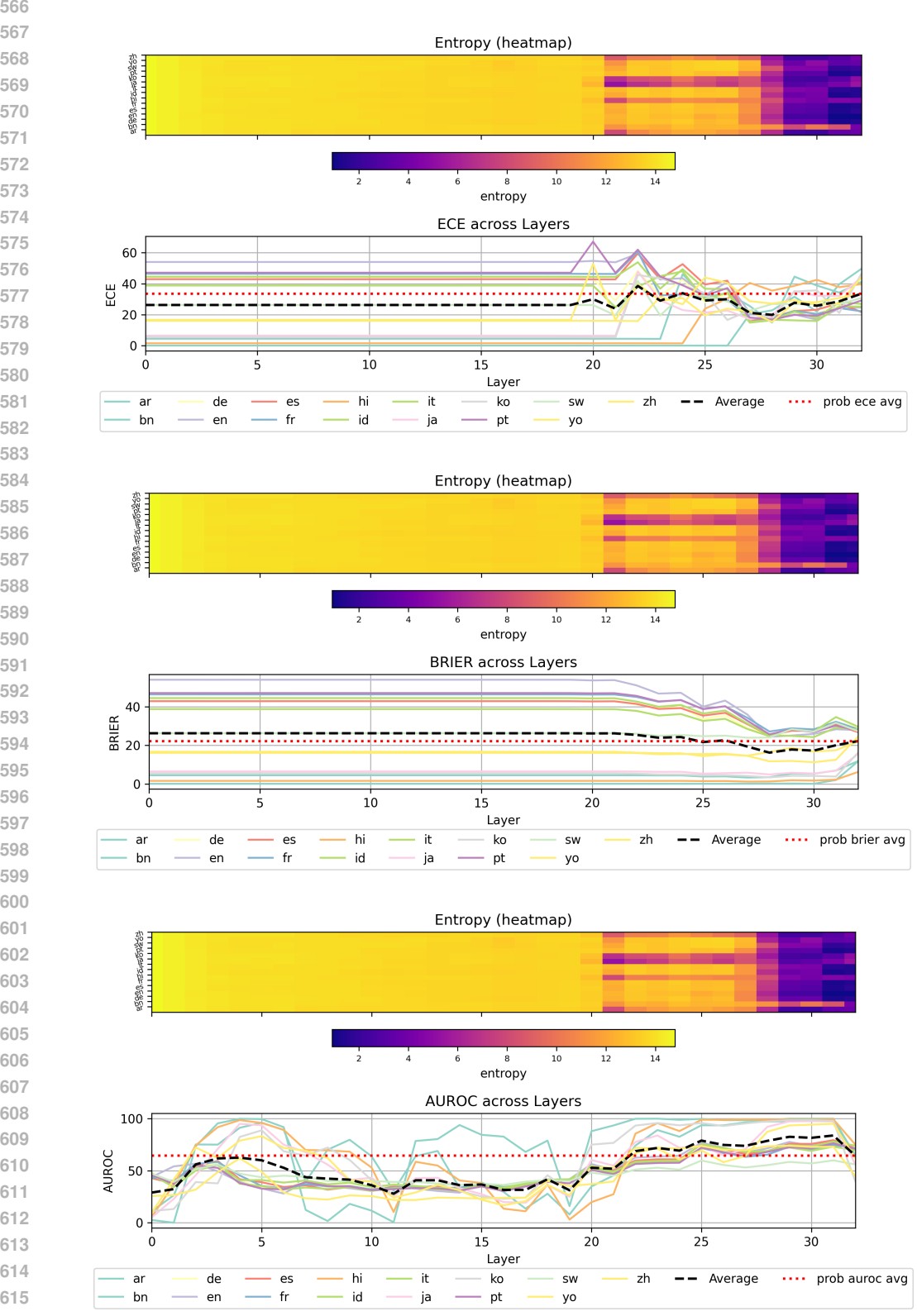

Figure 12: Calibration metrics (ECE, Brier score, AUROC) vs. entropy across layers on the MMMLU dataset for Mistral.

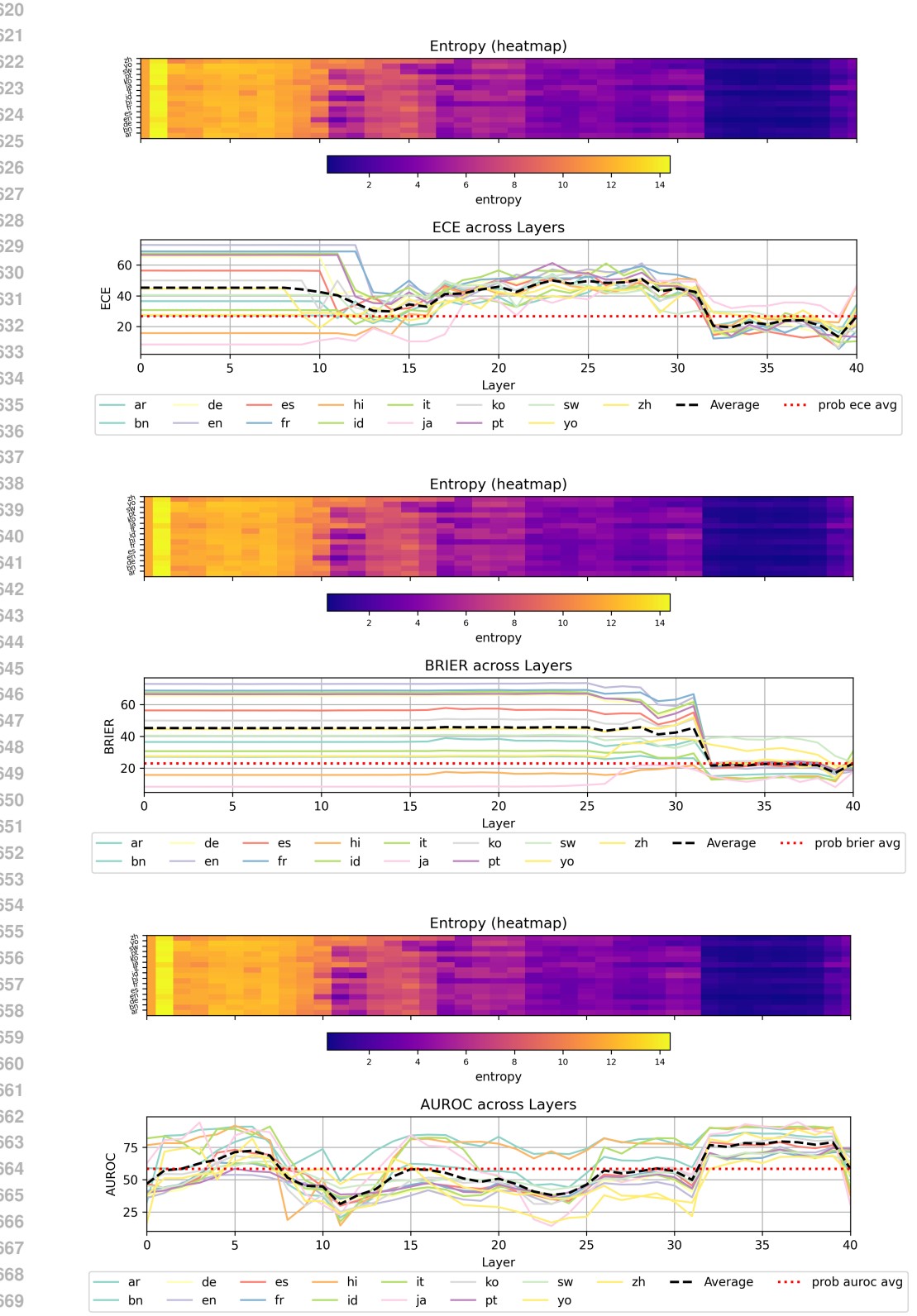

Figure 13: Calibration metrics (ECE, Brier score, AUROC) vs. entropy across layers on the MMMLU dataset for Phi.

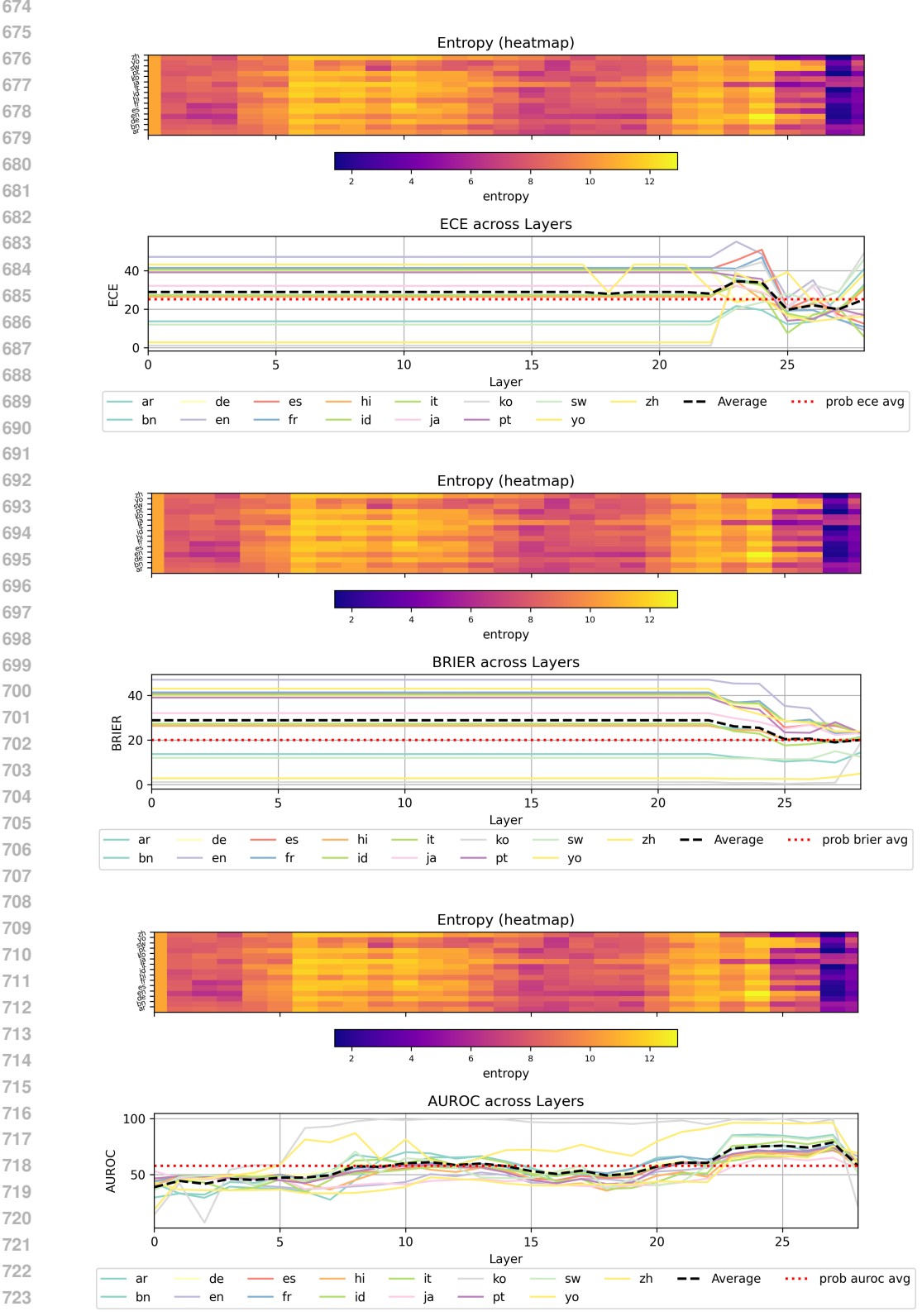

Figure 14: Calibration metrics (ECE, Brier score, AUROC) vs. entropy across layers on the MMMLU dataset for Deepseek.

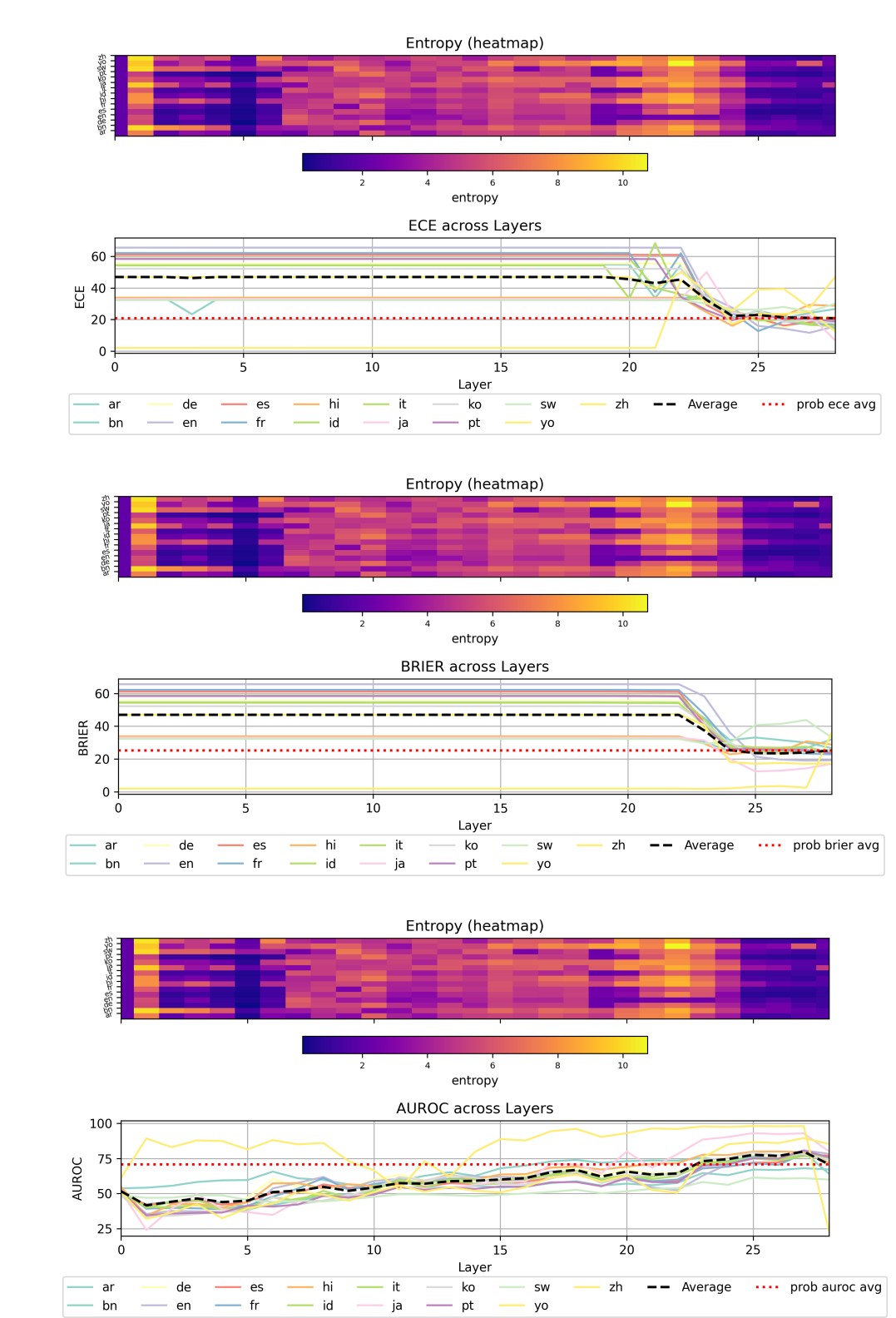

Figure 15: Calibration metrics (ECE, Brier score, AUROC) vs. entropy across layers on the MMMLU dataset for Qwen 2.5.

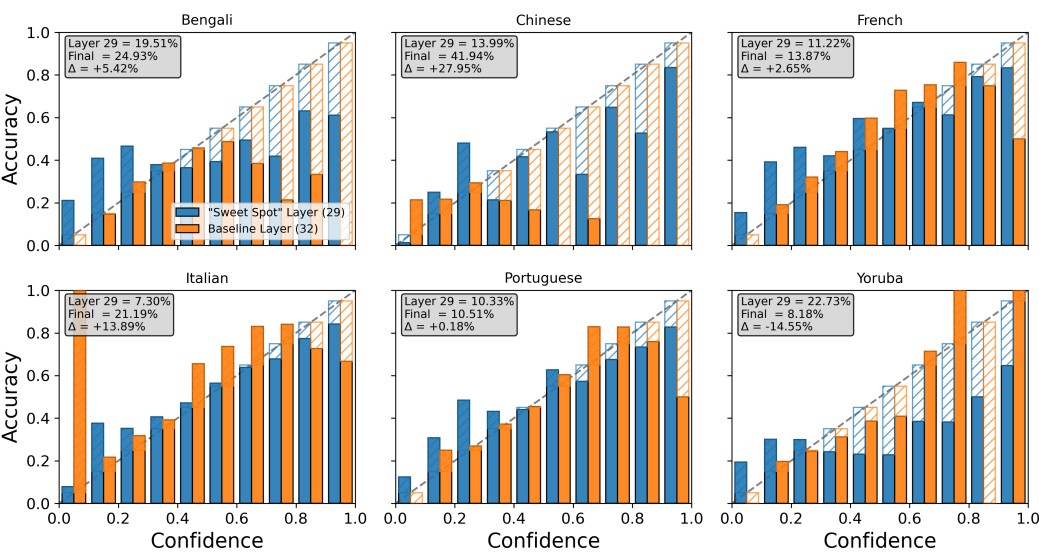

Figure 16: showing calibration curves for the remaining languages in LLaMA3: Bengali, Chinese, French, Italian, Portuguese, and Yoruba. Each plot compares the "sweet spot" intermediate layer (Layer 29, blue) against the final layer (Layer 32, orange). Bars represent accuracy across confidence bins, with diagonal dashed lines indicating perfect calibration. Reported values denote ECE for Layer 29 and the final layer, along with the relative improvement (Δ).

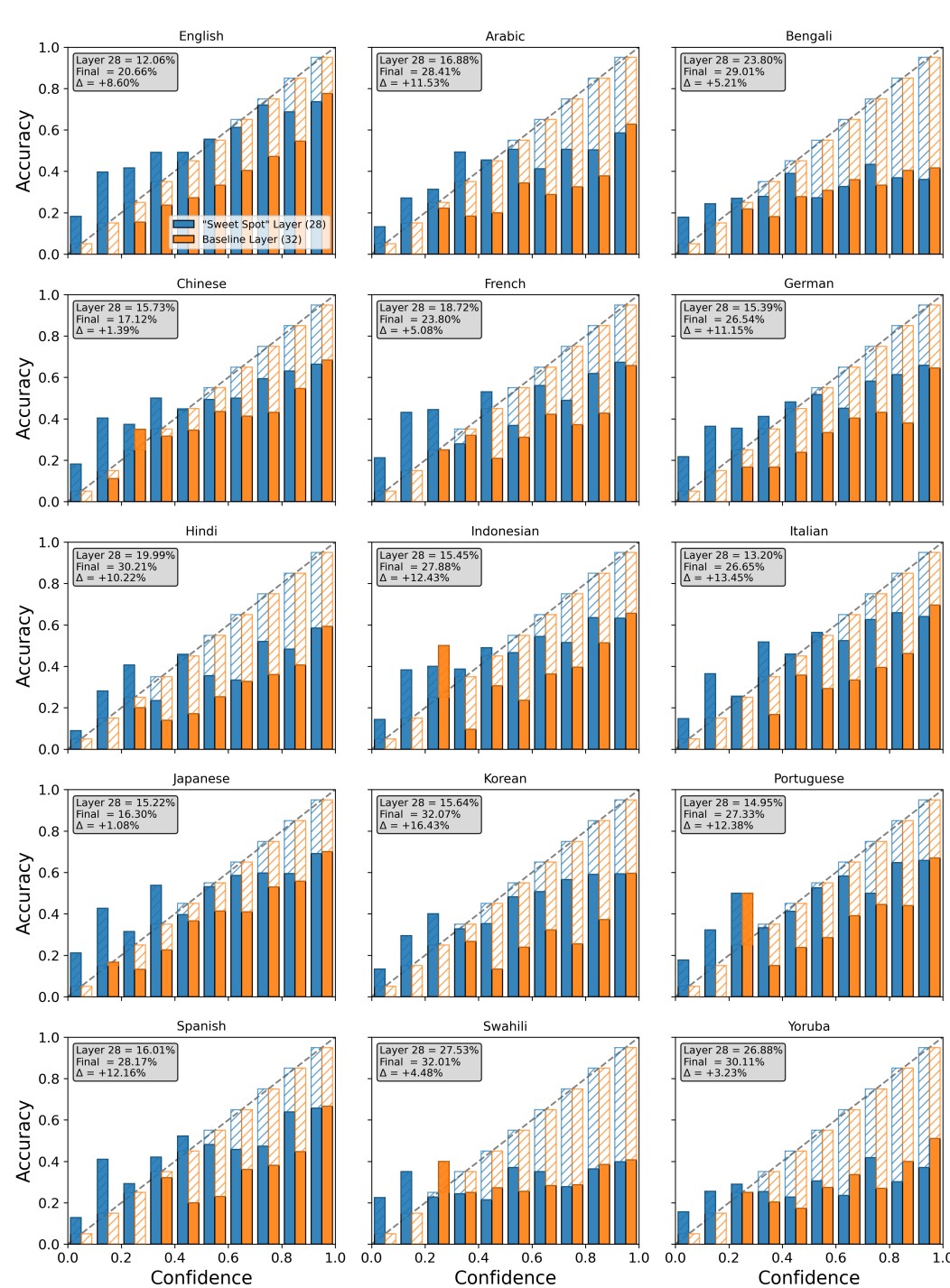

Figure 17: Reliability diagrams for Aya across all evaluated languages. Each plot compares the "sweet spot" intermediate layer with the final layer. Blue bars represent accuracy across confidence bins for the intermediate layer, while orange bars represent the final layer. The diagonal dashed line indicates perfect calibration. Reported values denote the ECE for Layer 28 and final layer, along with the relative improvement (Δ), highlighting how calibration changes across layers in different languages.

## D  POST-HOC CALIBRATORS

We include two widely used post-hoc calibration methods as baselines: **Temperature Scaling** (Guo et al., 2017) and **Isotonic Regression** (Zadrozny & Elkan, 2002). Both are trained on a held-out validation set (15k examples from MMMLU and 12k examples from Belebele, non-overlapping with the evaluation sets). The fitted calibrators are then applied to test-set predictions, and metrics (ECE, AUROC, Brier score, Accuracy) are computed using the calibrated probabilities.

**Temperature Scaling**  Temperature scaling applies a single scalar parameter $T > 0$ to rescale logits before computing probabilities. The parameter is optimized by minimizing the **negative log-likelihood (NLL)** on the validation set. In practice, we perform a coarse-to-fine grid search: first over a wide range ($T \in [0.05, 5.0]$ with 60 candidates), then locally refining around the best value with a denser grid. This procedure provides stable estimates across languages and avoids degenerate minima. The resulting optimal temperature is then used to rescale logits of all models prior to evaluation.

**Isotonic Regression**  Isotonic regression learns a non-parametric, monotone mapping from predicted probabilities to calibrated probabilities in $[0, 1]$. We use the `scikit-learn` implementation with out-of-bounds clipping and monotonicity constraints. The model is fitted on the validation set and then applied to test-set predictions.

# E ADDITIONAL ANALYSES

## E.1 EXPERIMENTS ON THE 70B MODEL

To answer whether the calibration patterns observed in 7B–8B models persist in much larger models, we conducted a layer-wise calibration study on the **LLaMA-3.1 70B Instruct** model, using the same multilingual MMMLU setup.

**Experimental Setup.** We evaluated the 70B model under identical conditions as the 8B model: 15 languages and 15,000 test examples. To manage computational demands, the model was loaded using 4-bit quantization, which reduces memory usage while preserving performance. All experiments were performed on NVIDIA H100 80GB GPUs. LLaMA-3.1 70B has **80 transformer layers** (compared to 32 in the smaller models).

**Layer-wise Calibration in a Large Model.** The results, shown in Figures 18, 19, and 20, clearly demonstrate that the phenomenon observed in smaller models is also present in the 70B model: Calibration improves substantially in the late-intermediate layers, well before the final layer.

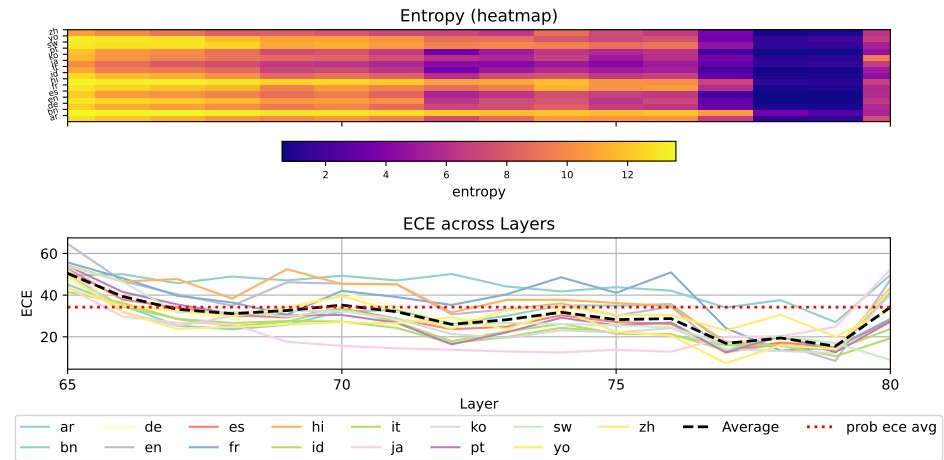

Figure 18: Expected Calibration Error (ECE) across the 80 layers of LLaMA-3.1 70B. A clear calibration minimum is visible in the intermediate layers across all language groups.

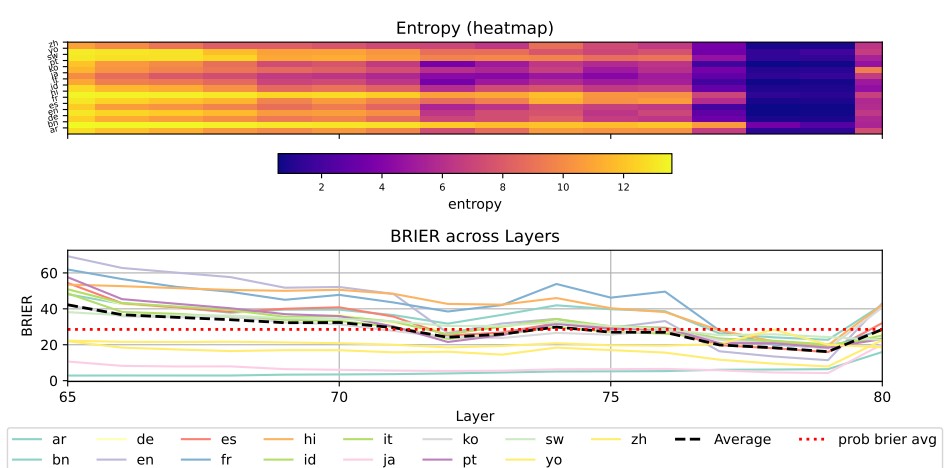

Figure 19: Brier score across the 80 layers of LLaMA-3.1 70B. The best-calibrated layers occur before the final layer, mirroring the pattern seen in ECE.

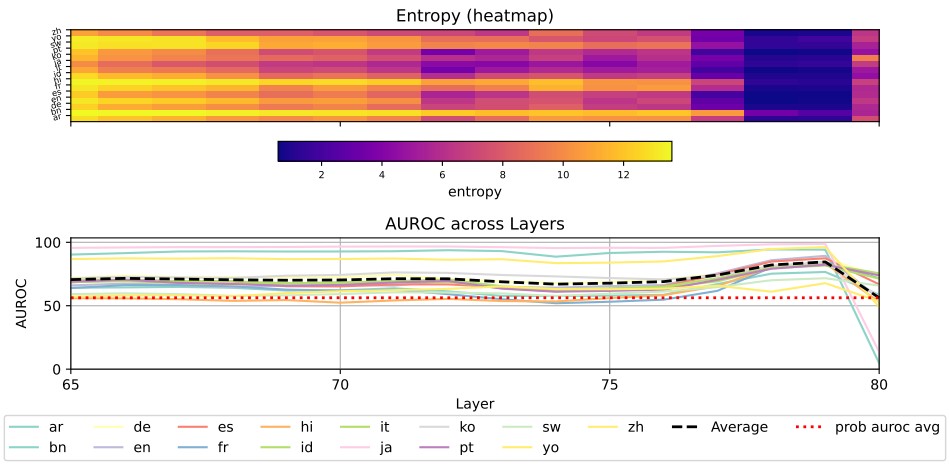

Figure 20: AUROC across the 80 layers of LLaMA-3.1 70B. The improvement in discriminative performance at intermediate layers is consistent with the trends in smaller models.

**Quantifying the Improvement.** To evaluate whether our calibration methods continue to be effective at larger scales, we also applied BEST LAYER, GOOD LAYER, and LACE to the 70B model. The results are reported in Table 14. Following the same format as Table 2 in the main text, this table shows that the intermediate-layer signal remains strong in the 70B model: using probabilities from the best-performing intermediate layer yields substantially better calibration than relying on the final layer. Moreover, LACE continues to provide the largest overall improvement, achieving the best calibration among all methods. These results confirm that our proposed layer-based calibration strategies, including LACE, scale effectively to much larger models.

| Method | ECE | Brier | AUROC |
|---|---|---|---|
| ORIGINAL PROBABILITY[†] | 23.51 | 22.10 | 65.19 |
| *(+Temperature Scaling)* | 22.56 | 21.97 | 65.19 |
| *(+Isotonic Regression)* | 19.01 | 21.34 | 65.07 |
| BEST LAYER[†] | 16.31 | 16.14 | 84.57 |
| *(+Temperature Scaling)* | 13.81 | 14.94 | 84.57 |
| *(+Isotonic Regression)* | 7.62 | 13.77 | 84.40 |
| GOOD LAYER[†] | 13.90 | 16.43 | 79.30 |
| *(+Temperature Scaling)* | 15.00 | 16.42 | 79.30 |
| *(+Isotonic Regression)* | 13.44 | 16.25 | 78.98 |
| LACE[‡] | 6.14 | 15.76 | 79.42 |
| *(+Temperature Scaling)* | 4.93 | 15.60 | 79.92 |
| *(+Isotonic Regression)* | 2.81 | 15.38 | 78.79 |

Table 14: Calibration performance of different layer-based methods on the LLaMA-3.1 70B model using MMMLU. Intermediate-layer methods continue to yield substantial calibration improvements even at large model scale.

## E.2 ECE BINNING SENSITIVITY

In the main paper, following Guo et al. (2017), we report ECE using $K = 10$ uniformly spaced bins over $[0, 1]$, applying the identical binning scheme across all models, layers, languages, and calibration methods.

To evaluate robustness with respect to this hyperparameter, we recomputed ECE for MMMLU predictions for both LLaMA-3 and Aya using $K \in \{5, 10, 15, 20, 25\}$. Table 15 presents the complete results for both LLaMA-3 and Aya, while Figure 21 visualizes the per-layer trends for LLaMA-3.

| Bins (K) | LLaMA-3 ECE (%) | | Aya ECE (%) | |
|---|---|---|---|---|
| | English | Non-English | English | Non-English |
| 5 | 3.21 | 17.87 | 19.90 | 25.93 |
| 10 | 4.61 | 23.12 | 20.66 | 26.77 |
| 15 | 5.04 | 23.27 | 20.34 | 27.45 |
| 20 | 5.58 | 22.83 | 21.72 | 28.26 |
| 25 | 6.07 | 23.13 | 21.34 | 28.27 |

Table 15: ECE binning sensitivity analysis for LLaMA-3 and Aya on MMMLU. English consistently shows lower calibration error than non-English across all binning configurations.

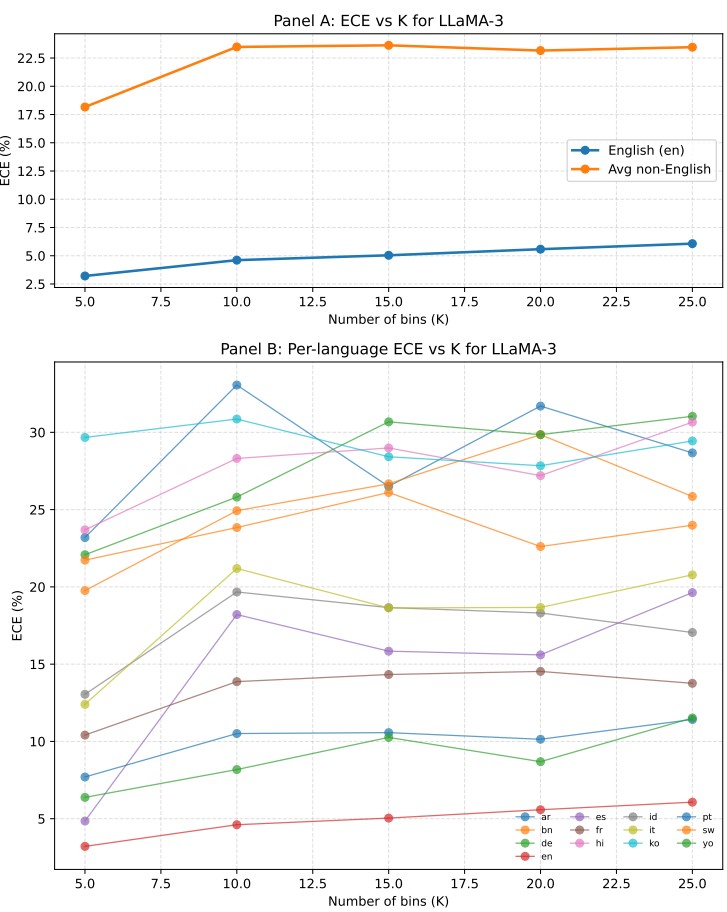

Figure 21: Per-layer ECE for LLaMA-3 on MMMLU across different numbers of bins $K$. The calibration gap between English and non-English languages persists across all binning configurations.

Although absolute values shift somewhat under different bin counts, the *qualitative pattern remains unchanged* across all settings:

- Non-English languages remain substantially less calibrated than English for every choice of $K$.
- Non-English calibration is highly stable once $K \geq 10$: LLaMA-3 stays within a narrow 22.83–23.27% band, and Aya within 26.77–28.27%.

The results confirm that the observed multilingual calibration gap holds with different binning choices. Furthermore, both the Brier score and AUROC (*bin-free*) exhibit the same cross-lingual trends, therefore the consistency across bin settings and metrics demonstrates that our conclusions are robust to reasonable variations in the ECE computation scheme.

### E.3 SIMULATION OF LANGUAGE-ID NOISE FOR LACE

To quantify how sensitive LACE is to noisy language labels, we run a controlled corruption experiment on MMMLU for two base models (Aya and LLaMA-3).

We start from our setup in the main paper where language-specific layer sets are selected on the validation split using gold language labels. Then at evaluation time, we simulate language-ID mistakes by randomly corrupting the language tag for a proportion $p \in \{0.0, 0.1, 0.2\}$ of test examples: for each selected example, we replace its true language label with a *different* language sampled from the remaining languages. LACE uses these (possibly incorrect) language tags to choose which language-specific layer set to average, while the underlying model logits remain unchanged.

Therefore, $p = 0$ corresponds to perfect language identification, whereas $p = 0.1$ and $p = 0.2$ approximate moderate error rates in an external language-ID system. We compute ECE, AUROC, and Brier score for each corruption level. The results are shown in Table 16.

| Corruption | Model | ECE (%) | AUROC (%) | Brier (%) |
|---|---|---|---|---|
| 0% | LLaMA-3 | 5.96 | 72.94 | 19.91 |
| | Aya | 11.42 | 68.61 | 23.16 |
| 10% | LLaMA-3 | 5.97 | 72.80 | 19.97 |
| | Aya | 11.60 | 68.38 | 23.31 |
| 20% | LLaMA-3 | 6.29 | 72.39 | 20.15 |
| | Aya | 12.07 | 68.25 | 23.39 |

Table 16: Robustness of LACE to language-ID corruption on MMMLU.

The impact of language-ID noise is mild. For Aya, corrupting 10% of tags increases ECE by only $0.18$ (about 1.5% relative) and Brier by $0.14$, while AUROC decreases by $0.23$. Even at 20% corruption, ECE rises by only $0.65$ compared to the clean setting, and Brier changes by $0.23$. For LLaMA-3, the effects are even smaller: ECE changes by $+0.01$ at 10% and $+0.33$ at 20%, with AUROC decreasing by less than $0.60$ and Brier increasing by at most $0.24$. These variations are negligible relative to the large cross-lingual calibration gaps reported in the main paper.

Intuitively, LACE remains robust because it averages over multiple "good" layers per language rather than relying on a single, highly language-specific layer: misrouting an example to another language's layer set still yields a reasonable ensemble, so moderate language-ID noise only weakly perturbs the final predictions.

### E.4 CROSS-LINGUAL LAYER-SHARING AND TRANSFER EFFECTS

To better understand whether calibration improvements transfer across languages, we analyze the extent to which languages share similar "good layer" sets under our intermediate-layer calibration framework. If two languages select highly overlapping sets of good layers, we would expect stronger cross-lingual transfer: optimizing for one language could benefit the other. Conversely, disjoint sets would suggest that calibration signals are language-specific.

**Measuring cross-lingual similarity.** For each language $\ell$, let $G_\ell$ denote its selected set of good layers. We compute a language–language similarity matrix using the Jaccard index:

$$\text{Jaccard}(\ell_i, \ell_j) = \frac{|G_{\ell_i} \cap G_{\ell_j}|}{|G_{\ell_i} \cup G_{\ell_j}|}.$$

A score of $1.0$ indicates identical layer sets, while $0.0$ means no overlap. Higher similarity reflects stronger potential cross-lingual transfer.

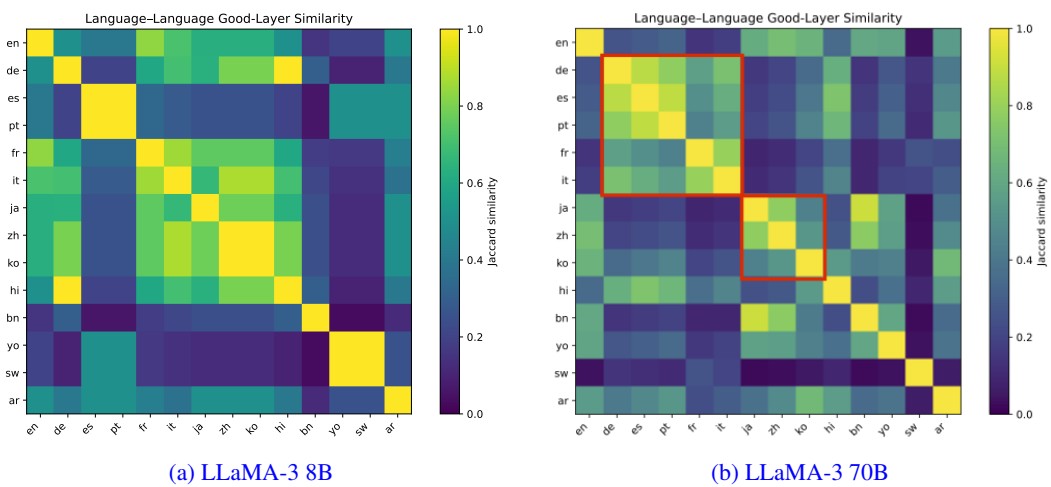

(a) LLaMA-3 8B  (b) LLaMA-3 70B

Figure 22: Cross-lingual Jaccard similarity between languages' good-layer sets. Clear correlation between Western European and CJK language groups indicate stronger organization of calibration-relevant information in deeper layers.

**Findings.** Figure 22 shows the cross-lingual similarity matrices for LLaMA-3 8B and 70B. While both models show some alignment among typologically related languages, the 70B model exhibits *far clearer and more coherent clustering*, reflecting stronger structural organization in its intermediate representations. In particular:

- **Western European languages** (e.g., German, Spanish, Portuguese, French, Italian) share overlapping good-layer subsets.
- **CJK languages** (Japanese, Chinese, Korean) also form a consistent cluster.
- Languages across distant families (e.g., English vs. CJK; Romance vs. Indic or African languages) show much lower overlap.

Example good-layer sets within Romance languages in the 70B model illustrate this structured overlap:

- `de`: [72, 75, 76, 77, 78, 79, 80]
- `es`: [72, 73, 75, 76, 77, 78, 79, 80]
- `pt`: [71, 72, 73, 75, 76, 77, 78, 79, 80]
- `fr`: [77, 78, 79, 80]
- `it`: [72, 77, 78, 79, 80]

Overall, these results confirm that **similar languages tend to share partially overlapping calibration-relevant layer patterns**, while more distant languages exhibit far less similarity.

### E.5 CROSS-DATASET CONSISTENCY OF GOOD-LAYER SELECTION

We assess whether "good layers" for calibration are consistent across different tasks by performing a cross-dataset layer-overlap analysis. For each language $\ell$ common to both MMMLU and Belebele, we extract the good-layer sets $G^{\text{MMMLU}}$ and $G^{\text{Belebele}}$ and compute their Jaccard similarity:

$$\text{Jaccard}(m) = \frac{1}{|\mathcal{L}_m|} \sum_{\ell \in \mathcal{L}_m} \frac{\left| G_{m,\ell}^{\text{MMMLU}} \cap G_{m,\ell}^{\text{Belebele}} \right|}{\left| G_{m,\ell}^{\text{MMMLU}} \cup G_{m,\ell}^{\text{Belebele}} \right|}.$$

This evaluation was conducted across the 12 languages common to both datasets. The results show moderate cross-dataset consistency, with an average Jaccard similarity of **0.5227** for LLaMA-3 and **0.4704** for Aya. This indicates that despite the different task formulations, the benchmarks identify partially overlapping sets of calibration-relevant layers. This suggests that certain structural properties of intermediate representations support robust calibration across tasks, though task-specific effects remain.

**Transfer evaluation on MMMLU using Belebele-derived layers.** To further test generalization, we applied the good-layer sets identified on Belebele directly to MMMLU without any post-hoc calibration. The results, shown in table 17, confirm that Belebele-derived layers retain meaningful calibration performance on MMMLU, demonstrating clear cross-task transferability.

| Model | Setting | ECE $\downarrow$ | AUROC $\uparrow$ | Brier $\downarrow$ |
|-------|---------|------|-------|-------|
| LLaMA-3 | baseline | 22.2 | 66.1 | 23.3 |
|  | *transferred*-LACE | **15.6** | **70.6** | **21.8** |
| Aya | baseline | 27.4 | 68.9 | 32.3 |
|  | *transferred*-LACE | **19.8** | **69.3** | **25.2** |

Table 17: Calibration transfer results when applying **Belebele-derived good layers** directly to **MMMLU** (without post-hoc calibration).

