# OpenReview forum: "Beyond the Final Layer: Intermediate Representations for Better Multilingual Calibration in Large Language Models"
_ICLR.cc/2026/Conference — Submitted to ICLR 2026_

### Official Review · Reviewer_UaFf · 2025-10-24

**Soundness:** 3
**Presentation:** 4
**Contribution:** 3
**Rating:** 6
**Confidence:** 3

**Summary:**

This paper presents a comprehensive study on the calibration of LLMs, revealing that non-English languages suffer from poor calibration. A layer-by-layer analysis indicates that this issue stems from an English bias in the final layer, while intermediate layers closer to the output provide better-calibrated signals for other languages. The authors introduce LACE, a training-free approach that combines these optimal intermediate layers for each language. Experiments on MMMLU and Belebele demonstrate that LACE significantly improves calibration accuracy and enhances existing post-hoc methods.

**Strengths:**

1. This paper tackles the critical issue of multilingual LLM reliability at a landmark scale (6 models, 100+ languages) using rigorous, human-curated data.
2. The main finding is both clear and actionable: intermediate layers are better calibrated for non-English languages, which provides a concrete explanation for the observed language bias.
3. The experimental design is solid.
4. The proposed LACE method is practical, effective, and works with existing post-hoc calibrators, making it highly valuable.

**Weaknesses:**

1. The scope is limited to multiple-choice questions. We don't know if this intermediate-layer advantage holds for open-ended generation, which is a significant gap given how LLMs are actually used.
2. LACE requires labeled validation sets for each language, which could be a significant barrier for low-resource languages that need this method the most. The data requirement may limit the widespread applicability of the solution in real-world settings.
3. The "English-bias" hypothesis doesn't fully explain why "multilingual-first" models like Aya also show better intermediate-layer calibration.

**Questions:**

1. Could you speculate on a path forward for testing this hypothesis on generative tasks (e.g., correlating layer-wise probabilities with human judgments)?

---

> ### Author Response · Authors · 2025-11-20
>
> Thank you for your constructive feedback regarding the scope of our tasks, data requirements, and the interpretation of our hypothesis. We have addressed these points with clarifications and new experiments below.
>
> > The scope is limited to multiple-choice questions.
>
> Thank you for pointing this out. We focused on MCQA to ensure methodological rigor and internal validity. Our reasoning is twofold:
>
> - **Unambiguous Correctness for calibration calculation:** Open-ended generation introduces unexpected noise when evaluating the correctness of model responses. A recent study highlights a crucial pitfall: standard QA evaluation relies on 'approximate correctness functions' (like ROUGE, BLEU, or LLM-as-a-judge) that often disagree with one another. Because these metrics vary, conclusions can **even flip depending on which one is used**. Focusing on MCQA avoids this problem.
> - **Controllable Confidence Estimation:** In MCQA, confidence is precisely defined as the probability of the selected token. In contrast, confidence estimation in open-ended generation is an active research field with high variance (e.g., requiring verbalized confidence, perplexity aggregation, or sampling consistency). Using MCQA allows us to isolate the *layer's* contribution to calibration without the confounding variables introduced by complex generation-based confidence heuristics.
>
> [1] Addressing Pitfalls in the Evaluation of Uncertainty Estimation Methods for Natural Language Generation. https://arxiv.org/abs/2510.02279
>
> > Could you speculate on a path forward for testing this hypothesis on generative tasks?
>
> To extend our findings to open-ended generation (e.g., short-form QA), we propose the following roadmap:
>
> 1.  **Establishing Correctness:** We recommend using an **"LLM-as-a-judge"** framework to handle semantic variance in generated answers, ensuring robust evaluation of accuracy.
> 2.  **Confidence Estimation Methods:** We propose testing the sequence-level confidence estimation methods applied to intermediate layers: calculating the perplexity or Negative Log-Likelihood (NLL) for the sequence at each layer (closest to our current approach).
>
> We are happy to include this to the paper Limitation Section if needed.

---

> > ### Author Response · Authors · 2025-11-20
> >
> > > LACE requires labeled validation sets for each language, which could be a significant barrier for low-resource languages that need this method the most.
> >
> > In our experiments, we used only **1,000 datapoints** per language for validation. We believe this is a feasible amount of data to collect even for low-resource languages.
> >
> > To further address this concern, we conducted a sensitivity analysis reducing the validation set size to 500, 200, and 100 examples. We measured the Jaccard similarity of the selected "Good Layers" against the full 1,000-sample baseline.
> >
> > Even reducing the validation set to **500 examples** retains very high alignment with the full set.
> >
> > | Comparison (Samples) | Average Jaccard Similarity | Interpretation |
> > | :--- | :--- | :--- |
> > | **1000 vs 500** | **0.88** | **High Consistency** |
> > | 1000 vs 200 | 0.65 | Moderate Consistency |
> > | 1000 vs 100 | 0.51 | Lower Consistency |
> >
> > This suggests that LACE is robust and effective with as few as 500 labeled examples, significantly lowering the barrier to entry.
> >
> > > The "English-bias" hypothesis doesn't fully explain why "multilingual-first" models like Aya also show better intermediate-layer calibration.
> >
> >
> > We appreciate this nuance. Upon closer inspection, the "English-bias" hypothesis remains valid for the Aya model due to three key factors:
> >
> > 1. **The Gap Persists (Magnitude vs. Existence)**
> > While Aya reduces the *magnitude* of the disparity compared to LLaMA-3, the "English-bias" persists in the hierarchy of performance. As shown in **Table 1**, English consistently outperforms other languages in calibration across both models. For LLaMA-3, the gap is drastic, with English ECE at **4.61%** versus Hindi at **28.31%** and Arabic at **33.06%**. For Aya, although the gap is compressed, English (**20.66%**) still maintains a significant advantage over Hindi (**30.21%**) and Arabic (**28.41%**). The pattern remains English-centric despite the multilingual-first training.
> >
> > 2.  **Training Data Composition & Quality:** Although Aya is multilingual-first, English data still constitutes a significant portion (**21.5%**) of the training set and is categorized as "High-Resource" [2]. Crucially, the English instructions are original, human-written text, whereas instructions for 101 other languages were generated using machine translation. This results in higher coherence and quality for the English representations.
> >
> > 3.  **Model Initialization:** Aya is not trained from scratch; it is fine-tuned on **mT5**, which was pre-trained on **mC4** (1 trillion tokens of web text). Web-crawled corpora like mC4 are historically dominated by English in terms of volume and domain diversity. Consequently, the "base" model possesses a stronger latent understanding of English before instruction fine-tuning even begins, which persists into the intermediate layers.
> >
> > **Reference:**
> >
> > [1] SelfCheckGPT: Zero-Resource Black-Box Hallucination Detection for Generative Large Language Models, EMNLP 2023
> >
> > [2] Aya Model: An Instruction Finetuned Open-Access Multilingual Language Model, ACL 2024.
> >
> > ---
> >
> > We hope these additional analyses can further strengthen our paper. Thank you again for raising these constructive points.

---

> > > ### Author Response · Authors · 2025-11-28
> > >
> > > Dear Reviewer UaFf,
> > >
> > > Thank you for your constructive feedback. We have updated the PDF accordingly, and the revised version now includes the requested clarifications and new analyses. Please let us know if any further issues remain.
> > >
> > > Best regards,
> > >
> > > The Authors

---

### Official Review · Reviewer_aNHy · 2025-10-31

**Soundness:** 3
**Presentation:** 3
**Contribution:** 3
**Rating:** 6
**Confidence:** 2

**Summary:**

The paper "Beyond the Final Layer: Intermediate Representations for Better Multilingual Calibration in Large Language Models" investigates the calibration of multilingual large language models (LLMs) and highlights the disparities in performance between English and non-English languages. Overall, the study emphasizes the importance of considering intermediate representations for enhancing the reliability of multilingual LLMs.

**Strengths:**

1. The paper introduces a fresh perspective on the calibration of multilingual LLMs by emphasizing the importance of intermediate layers rather than solely relying on the final layer for confidence estimation. This approach challenges established norms in the field and provides a new framework for understanding how different layers contribute to model performance.

2. The research is grounded in a comprehensive empirical analysis that utilizes high-quality datasets spanning over 100 languages. The methodology is rigorously designed, employing well-defined metrics to evaluate model performance.

**Weaknesses:**

1. LACE is a very interesting approach. I’m curious whether optimizing for a specific language benefits similar languages. For instance, if we optimize for Japanese, would it improve calibration for Korean or negatively impact English understanding? Exploring this could shed light on cross-lingual transfer effects.

2. The LACE experiment is conducted within a single dataset. I wonder if optimizing LACE on one dataset could enhance model performance on another for a specific language. This would help determine whether the method generalizes across datasets rather than being dataset-specific.

3. As mentioned in the limitations, all evaluated models are relatively small. It remains unclear whether the conclusions hold for larger models (e.g., >30B parameters). Investigating this would strengthen the findings.

**Questions:**

See above

---

> ### Author Response · Authors · 2025-11-20
>
> Thank you for your positive assessment of LACE and for raising three critical questions regarding transferability and scaling. We have conducted additional experiments to address each of these points, with full details provided in the new **Appendix E** of the revised paper.
>
>
> > 1. Curious whether optimizing for a specific language benefits similar languages (cross-lingual transfer). (Appendix E.4)
>
> This is indeed a very interesting direction to explore. We investigated the potential for cross-lingual transfer by analyzing the overlap of "good layers" across languages. As shown in **Figure 20** (Appedinx E.4) in the revised paper, we found that typologically similar languages (e.g., Romance languages like Spanish, French, Italian; and CJK languages) share highly overlapping sets of optimal calibration layers. This confirms that optimizing for a source language can indeed benefit similar target languages, whereas distant languages (e.g., English vs. CJK) exhibit distinct calibration profiles.
>
> > 2. Wonder if optimizing LACE on one dataset could enhance performance on another (cross-dataset generalization). (Appendix E.5)
>
> To test dataset independence, we first calculated the "good layers" sets on both **Belebele** and **MMMLU** dataset and found Jaccard Similarity of 0.5227 for LLaMA-3 and 0.4704 for Aya. We then applied the layer parameters learned on the **Belebele** dataset directly to the **MMMLU** benchmark without further modification. As detailed in **Table 17**, the method generalizes robustly: applying Belebele-derived layers to MMMLU reduced the ECE on LLaMA-3 from 22.2% (baseline) to 15.6%. This demonstrates that the "calibration sweet spot" is an architectural property of the model rather than an artifact of a specific dataset.
>
>
> > 3. Unclear whether the conclusions hold for larger models (e.g., >30B parameters). (Appendix E.1)
>
> We extended our analysis to **LLaMA-3.1 70B** to verify if our findings hold at scale. The results (visualized in **Figures 16–18** and quantified in **Table 14**) confirm that the phenomenon persists: calibration quality improves significantly in the late-intermediate layers of the 70B model. Furthermore, LACE remains the most effective method, reducing ECE to **2.81%** (with isotonic regression) compared to the baseline.
>
> ---
>
>
> We hope these additional analyses can further support the robustness and generalizability of our paper. Thank you again for raising these constructive points.

---

> > ### Author Response · Authors · 2025-11-28
> >
> > Dear Reviewer aNHy,
> >
> > Thank you for your positive assessment and for highlighting the three key questions. We have now updated the PDF, and the revised version includes all requested analyses: cross-lingual transfer (App. E.4), cross-dataset generalization (App. E.5), and scaling to larger models (App. E.1). Please let us know if any further clarification is needed.
> >
> > Best regards,
> >
> > The Authors

---

> > > ### Comment · Reviewer_aNHy · 2025-11-28
> > >
> > > Thanks for your detailed and extensive explanation, I will raise my score.

---

### Official Review · Reviewer_AGvT · 2025-11-06

**Soundness:** 2
**Presentation:** 3
**Contribution:** 2
**Rating:** 4
**Confidence:** 3

**Summary:**

The paper conducts a large-scale benchmarking study of multilingual calibration in LLMs (six model families; 100+ languages) on MMMLU and Belebele, documenting a substantial English vs. non-English calibration gap and a mid-layer “sweet spot” where ECE improves for many non-English languages. Building on this descriptive finding, the authors propose training-free confidence extractors (Best Layer, Good-Layers Ensemble) and LACE (a language-aware ensemble), sometimes combined with post-hoc calibration (temperature / isotonic), which reduce ECE on these benchmarks.

**Strengths:**

Timely and important problem: Multilingual calibration is underexplored and high-impact for safety/trust. The English vs. non-English gap is clearly quantified across models/datasets.

Simple, training-free methods: Best-Layer / Good-Layers / LACE are conceptually simple, require no retraining, and still deliver large ECE reductions, while remaining complementary to standard post-hoc calibration.

**Weaknesses:**

Limited ablation/depth of analysis. I could not find ablations on: (i) ECE binning sensitivity, (ii) computing calibration using each layer’s own prediction vs. probability on the final prediction, (iii) robustness to language-ID errors / code-switching (needed for LACE), and (iv) runtime/compute overhead of multi-layer scoring. The paper’s “Benchmarking” focus and method section do not report these studies.

Narrow scope and stated limitations. Experiments are restricted to 7B–8B models and MCQA; the authors explicitly leave open-ended generation to future work. Methods are post-hoc rather than integrated into training. These limitations make it hard to generalize the descriptive findings beyond the evaluated setting.


Theoretical grounding is thin. The paper offers a plausible narrative (final layer English-biased; intermediate layers more language-agnostic) but no formal analysis or causal probes explaining why the mid-layer sweet spot emerges. The results remain mostly empirical/observational.

**Questions:**

Ablations: How sensitive are ECE gains to the number/strategy of bins?

Layer-own predictions: If ECE/Brier are computed using each layer’s own argmax, do the mid-layer gains persist?


Language-ID robustness: How does LACE behave under language detection errors or code-switching? Any language-agnostic variant?


Cost/latency: What is the runtime overhead of Good-Layers / LACE vs. final-layer baselines?

Beyond MCQA: Any evidence on open-ended tasks (long-form QA, summarization) to show the phenomenon carries over?

---

> ### Author Response · Authors · 2025-11-20
>
> Thank you for recognizing the timeliness and importance of our work, as well as the effectiveness of our proposed methods. We appreciate your constructive feedback regarding ablations and scope. We have addressed these points with additional experiments (including ablation analysis, scaling to 70B models and robustness tests), which are summarized below and detailed in the newly added *Appendix E*. These results will be reorganized and incorporated into the final camera-ready version.
>
>
> > **1: Limited ablation/depth of analysis.**
> > *(i) ECE binning sensitivity*
>
> We agree that ECE could be sensitive to binning parameters. To ensure our findings are robust, we (1) also utilized Brier Score and AUROC (which are threshold-independent), yielding consistent results (see Table 1 & 2 in the main paper), and (2) conducted a sensitivity analysis on bin numbers $K \in \{5, 10, 15, 20, 25\}$. As shown in the table below, while absolute ECE values fluctuate slightly, the relative gap (English vs. Non-English) remains stable.
>
>
> **Table: ECE Sensitivity to Bin Count ($K$)**
>
>
> | Metric | Model | K=5 | K=10 | K=15 | K=20 | K=25 |
> | :--- | :--- | :---: | :---: | :---: | :---: | :---: |
> | **English ECE** | LLaMA-3 | 3.21 | 4.61 | 5.04 | 5.58 | 6.07 |
> | **Avg Non-English ECE** | LLaMA-3 | 17.87 | 23.12 | 23.27 | 22.83 | 23.13 |
> | **English ECE** | Aya | 19.90 | 20.66 | 20.34 | 21.72 | 21.34 |
> | **Avg Non-English ECE** | Aya | 25.93 | 26.77 | 27.45 | 28.26 | 28.27 |
>
>
> A detailed description is given in Appendix E.2, with a visualization of per-language ECE change (Figure 19).
>
>
> > *(ii) Computing calibration using each layer’s own prediction vs. probability on the final prediction*
>
>
> Thank you for noting this. This is a crucial methodological distinction. We explicitly chose to calibrate confidence with respect to the **final layer's prediction** ($\hat{y}_L$).
>
>
> Our objective is to provide a reliability signal for the *deployed* model. In a standard inference setting, the user receives the answer generated by the final layer. Therefore, the confidence score must reflect the probability that the *final layer's* answer is correct. If we computed calibration based on layer $l$'s own prediction ($\hat{y}_l$), we would essentially be evaluating an "early-exit" model. While interesting, early layers often have lower accuracy. Our method specifically addresses the question: *"Given that the model outputs the answer $Y$, how confident is the intermediate layer that $Y$ is correct?"*
>
>
> > *(iii) Robustness to language-ID errors / code-switching*
>
>
> We respond to this point in two ways:
>
>
> 1.  **We do have language-agnostic variant:** The **Good Layers Ensemble** (Section 5.1) is inherently robust to unknown languages or code-switching. It aggregates signals from layers that perform well globally without requiring a specific language ID ($k$). As shown in Table 2, this method significantly outperforms the final layer baseline (e.g., reducing LLaMA-3 ECE from 22.44 to 11.84) and serves as the fallback for scenarios where Language ID is unavailable.
> 2.  **LACE Robustness:** LACE is empirically robust to Language-ID noise because it ensembles over a set of layers rather than selecting a single one. We performed a simulation where we randomly corrupted varying percentages of Language IDs (from 0% to 20%) in the validation split used for layer selection. As shown below, performance degradation is negligible even at 20% corruption.
>
>
> **Table: LACE Robustness to Language ID Corruption**
>
>
> | Corruption | Model | ECE | AUROC | Brier |
> | :---: | :---: | :---: | :---: | :---: |
> | **0%** | Cohere | 11.42 | 68.61 | 23.16 |
> | | LLaMA-3 | 5.96 | 72.94 | 19.91 |
> | **10%** | Cohere | 11.60 | 68.38 | 23.31 |
> | | LLaMA-3 | 5.97 | 72.80 | 19.97 |
> | **20%** | Cohere | 12.07 | 68.25 | 23.39 |
> | | LLaMA-3 | 6.29 | 72.39 | 20.15 |
>
>
> > *(iv) Runtime/compute overhead of multi-layer scoring*
>
>
> The runtime overhead of our method is **negligible**. Our method requires only a single forward pass (which is already required for generation). We simply extract logits from intermediate layers as the computation propagates. Unlike sampling-based consistency checks (which require multiple forward passes) or auxiliary models, our method adds no extra inference steps and virtually zero latency.

---

> > ### Author Response · Authors · 2025-11-20
> >
> > > **2: Narrow scope (Model Size, MCQA, Post-hoc).**
> >
> >
> > 1.  **Larger Models:** To address the concern regarding model scale, we have conducted additional experiments on **LLaMA-3-70B** (80 layers). We observe the same "sweet spot" phenomenon and calibration improvements, confirming that our findings generalize to larger parameter scales. These results have been added to the revised Appendix E.1: see Figure 16, 17, 18 for the visualization of the "sweet spot" phenomenon. We show our experiment with LACE on 70B in the table below. LACE still performs better than baselines and can be further boost by existing calibration methods.
> >
> > | Method | Calibration | ECE | Brier | AUROC |
> > | :---: | :---: | :---: | :---: | :---: |
> > | ORIGINAL PROBABILITY† | None | 23.51 | 22.10 | 65.19 |
> > |  | +Temperature Scaling | 22.56 | 21.97 | 65.19 |
> > |  | +Isotonic Regression | 19.01 | 21.34 | 65.07 |
> > | BEST LAYER† | None | 16.31 | 16.14 | 84.57 |
> > |  | +Temperature Scaling | 13.81 | 14.94 | 84.57 |
> > |  | +Isotonic Regression | 7.62 | 13.77 | 84.40 |
> > | GOOD LAYER† | None | 13.90 | 16.43 | 79.30 |
> > |  | +Temperature Scaling | 15.00 | 16.42 | 79.30 |
> > |  | +Isotonic Regression | 13.44 | 16.25 | 78.98 |
> > | LACE‡ | None | 6.14 | 15.76 | 79.42 |
> > |  | +Temperature Scaling | 4.93 | 15.60 | 79.92 |
> > |  | +Isotonic Regression | 2.81 | 15.38 | 78.70 |
> >
> >
> > 2. **MCQA vs. Generation:** We focused on MCQA for two primary methodological reasons:
> >   - **Unambiguous Correctness for calibration calculation:** Open-ended generation introduces unexpected noise when evaluating the correctness of model responses. A recent study highlights a crucial pitfall: standard QA evaluation relies on 'approximate correctness functions' (like ROUGE, BLEU, or LLM-as-a-judge) that often disagree with one another. Because these metrics vary, conclusions can **even flip depending on which one is used**. Focusing on MCQA avoids this problem.
> >  - **Controllable Confidence Estimation:** In MCQA, confidence is precisely defined as the probability of the selected token. In contrast, confidence estimation in open-ended generation is an active research field with high variance (e.g., requiring verbalized confidence, perplexity aggregation, or sampling consistency). Using MCQA allows us to isolate the *layer's* contribution to calibration without the confounding variables introduced by complex generation-based confidence heuristics.
> >
> > [1] Addressing Pitfalls in the Evaluation of Uncertainty Estimation Methods for Natural Language Generation. https://arxiv.org/abs/2510.02279
> >
> >
> > 3.  **Post-hoc Nature:** We view the post-hoc nature of our method as a **feature**, not a limitation. Retraining or fine-tuning LLMs is computationally prohibitive for most practitioners. LACE provides a training-free, "plug-and-play" solution that can be immediately applied to deployed models to improve safety without altering model weights.
> >
> >
> > > **3: Theoretical grounding is thin.**
> >
> > We clarify that proposing a new theoretical framework for transformer mechanics is outside the scope of this work. The theoretical mechanism explaining *why* intermediate layers are more language-agnostic is already the subject of extensive recent research (e.g., Bandarkar et al., 2024b; Wendler et al., 2024; Kojima et al., 2024)(line 98-107). We rely on these established insights rather than attempting to re-derive them.
> >
> >
> > Instead, our contribution is to identify and exploit a critical **gap** in how these properties apply to model calibration. Specifically:
> > 1.  **Novel Empirical Discovery:** While previous work analyzed representations for *accuracy* or *alignment*, we are the first to demonstrate that this "mid-layer neutrality" creates a systematic "sweet spot" for **calibration** across over 100 languages.
> > 2.  **Methodological Utility:** We translate this observation into actionable, training-free interventions (LACE) that significantly reduce calibration error without the need for expensive retraining.
> >
> >
> > Our value lies in the rigorous empirical verification of this phenomenon at scale ($10^5$ instances) and the practical methods derived from it, rather than in theoretical speculation about the internal causal mechanisms of transformers. We will explicitly list this into our Limitation Section.
> >
> > ----
> >
> > We hope these additional analyses and clarifications can further strengthen our paper. We would appreciate it if you could re-assess our paper in light of these new results.

---

> > > ### Comment · Reviewer_AGvT · 2025-11-26
> > > **Records of changes**
> > >
> > > Dear authors, could you please highlight your changes in the manuscript so that I can easily figure it out? Many thanks.

---

> > > > ### Author Response · Authors · 2025-11-26
> > > >
> > > > Dear Reviewer AGvT,
> > > >
> > > > All our updates are currently located in **Appendix E (pages 36–42 of the PDF)**. This placement is only provisional: we plan to re-organise the content into the main text or other appendix sections later. For now, we have kept everything together to ensure easy recognizability during the rebuttal period.
> > > > Would this be acceptable?

---

> > > > > ### Comment · Reviewer_AGvT · 2025-11-27
> > > > > **Additional page for rebuttal**
> > > > >
> > > > > Dear authors, you have an additional page during the rebuttal period. So try to put the revision in the main context so that I could evaluate it as a single manuscript. Many thanks

---

> ### Author Response · Authors · 2025-11-27
>
> Dear Reviewer,
>
> To summarize, here is a **TL;DR** version of our above rebuttal:
>
> * **1 (Robustness & Ablations):** We confirm the robustness of our results through:
>     * additional sensitivity analyses on ECE binning (showing stable relative gaps),
>     * justification for using final-layer predictions to ensure deployment relevance,
>     * stress tests showing LACE is robust to up to 20% language-ID corruption,
>     * and clarification that the runtime overhead is negligible (single forward pass).
> * **2 (Scope & Model Scale):** We address concerns regarding scope by:
>     * demonstrating that our findings and the "sweet spot" phenomenon generalize to larger models (LLaMA-3-70B),
>     * explaining our focus on MCQA to ensure unambiguous correctness evaluation compared to open-ended generation,
>     * and highlighting that the post-hoc nature is a "plug-and-play" feature that requires no retraining.
> * **3 (Theoretical Grounding):** We clarify our theoretical positioning by:
>     * distinguishing our contribution as the novel empirical application of established "mid-layer neutrality" theories,
>     * and emphasizing the value of rigorously verifying this phenomenon across 100+ languages.
>
> **We are incorporating the suggested changes and will inform you once the revised manuscript is uploaded.**
>
>
> Best,
>
> The authors

---

> > ### Author Response · Authors · 2025-11-28
> >
> > Dear Reviewer AGvT,
> >
> > Thank you for your continued attention to our submission. The updated PDF is now ready for your review. Please let us know if you have any further concerns or suggestions.
> >
> > Best regards,
> >
> > The Authors

---

### Author Response · Authors · 2025-12-03

Dear Area Chair,

As the discussion period draws to a close, we would like to provide a brief summary of our rebuttal and the improvements made to the paper:

1. **General Updates:** We have conducted extensive additional experiments to address reviewer feedback, including scaling our analysis to 70B parameter models, adding sensitivity analyses for calibration binning, and testing cross-lingual/cross-dataset transferability. These have been incorporated into the revised PDF.

2. **Reviewer AGvT (Score: 4):** We engaged in a detailed discussion regarding ablations and scope. The reviewer requested that we integrate our new experimental results and revisions directly into the manuscript, which we have completed.

3. **Reviewer aNHy (Score: 6):** We addressed this reviewer's questions regarding cross-lingual transfer and model scaling. The reviewer expressed satisfaction with our "detailed and extensive explanation" and **explicitly stated they will raise their score**.

4. **Reviewer UaFf (Score: 6):** While we have not received a follow-up response, we have fully addressed their concerns regarding data requirements (demonstrating LACE is effective with as few as 500 examples) and provided the requested clarifications on our task scope and hypotheses.

Thank you for your time and supervision of the review process.

Best regards,

The Authors

---

### Meta-Review · Area_Chair_ZauG · 2026-01-06

**Summary:**

* Limited technical novelty: The paper is heavily empirical, leading to concerns that it serves more as a benchmark (Position paper) or diagnostic study rather than a fundamental algorithmic advancement. The core contribution (LACE) is an incremental ensemble approach rather than a transformative architecture or training paradigm.

* Mechanistic gaps: While the paper recognizes that intermediate layers achieve better, it lacks a deep mechanistic explanation for the specific physicochemical or linguistic reasons why the final layer "poisons" non-English calibration.

* Inference overhead: The reliance on ensembling multiple intermediate layers raises concerns regarding increased inference latency, which could hinder deployment in real-time multilingual applications. Despite the training-free nature, the authors did not fully satisfy all reviewer concerns regarding the practical latency costs of ensembling representations during high-throughput inference.

* Generalizability across alignment types: It remains unclear if these findings hold for models aligned via methods other than standard RLHF, or if the "optimal layers" shift significantly as alignment techniques evolve.

**Reviewer Concerns:**

Overall, the paper received positive feedback; however, due to limited technical novelty, it is more appropriately categorized as an extensive empirical study than a fundamental algorithmic advancement (mentioned clearly in the list of contributions by the authorsors). While the discovery of an "intermediate layer advantage" for multilingual calibration is a valuable diagnostic contribution, the proposed LACE method is essentially an ensemble of existing representations rather than a transformative new architecture.

**Reviewer Scores:**

*The reviewer AGvT suggested a rating of 4  with confidence of 3

*The reviewer UaFf gave a rating of 6 with confidence: 3

* The reviewer aNHy also evaluated this work with a score of 6, while having confidence of 2

---

### Decision · Program_Chairs · 2026-01-26

Reject